# Potential for risk reduction of chronic health conditions through lifestyle in childhood cancer survivors

Aron Onerup [1,2] ✉, Qi Liu [3], Shizue Izumi[4], Stephanie B. Dixon [1,5], Rebecca M. Howell[6], Matthew J. Ehrhardt [1,5], Lenat Joffe [7], Lucie M. Turcotte [8], Deo Kumar Srivastava [9], Tara M. Brinkman[10], Kevin C. Oeffinger [11], Claire Snyder[12], Paul C. Nathan[13], Yutaka Yasui [9], Gregory T. Armstrong[1] & Kirsten K. Ness [1]

Childhood cancer survivors are at high risk for treatment-related chronic health conditions. How much of this risk can be attributed to lifestyle is not known. In this study, we assess associations between lifestyle and a range of chronic health conditions and estimate lifestyle-specific population attributable fractions for chronic health conditions in survivors and compare them to those of radiotherapy and chemotherapy. Here we show that unhealthy lifestyle is associated with higher risk for subsequent hypertension, dyslipidemia, diabetes, heart attack, heart failure, valvular disease, joint replacement, anxiety, depression, and impaired physical and mental quality of life. Disease proportions attributed to unhealthy lifestyle exceed those of chemotherapy and radiotherapy for hypertension, diabetes, joint replacement, anxiety, depression, and impaired physical and mental quality of life. Unlike previous cancer treatment exposures, lifestyle can be modified. We need to further develop and implement effective lifestyle interventions in childhood cancer survivors, promoting healthy weight and physical activity.

Survival following childhood cancer currently exceeds 85% in the United States (US)[1]. This has led to an increasing population of childhood cancer survivors, with over 500,000 individuals currently living in the US and 300,000–500,000 residing in Europe[2,3]. This population is at increased risk for a number of treatment-related chronic health conditions (CHC), including metabolic syndrome, cardiovascular disease, musculoskeletal disorders, subsequent malignant neoplasms (SMNs), and symptoms of emotional distress, leading to a reduced health-related quality of life (QoL) and an elevated risk for premature death[4–8]. Identification of treatment-related risk factors for CHCs has led to the development of national and international evidence-based guidelines for risk-based organ-specific screening[9–11].

Research in the general population has demonstrated that several of these CHCs can be mitigated by adherence to a healthy lifestyle,

[1]Department of Epidemiology and Cancer Control, St Jude Children´s Research Hospital, Memphis, TN, USA. [2]Department of Pediatrics, Institute of Clinical Sciences, Sahlgrenska Academy, University of Gothenburg, Gothenburg, Sweden. [3]School of Public Health, University of Alberta, Edmonton, AB, Canada. [4]Faculty of Data Science, Shiga University, Hikone, Japan. [5]Department of Oncology, St Jude Children´s Research Hospital, Memphis, TN, USA. [6]Radiation Physics Department, The University of Texas at MD Anderson Cancer Center, Houston, TX, USA. [7]Department of Pediatrics, Cohen Children's Medical Center, Northwell Health, New Hyde Park, NY, USA. [8]Department of Pediatrics, University of Minnesota, Minneapolis, MN, USA. [9]Department of Biostatistics, St Jude Children´s Research Hospital, Memphis, TN, USA. [10]Department of Psychology and Biobehavioral Sciences, St Jude Children´s Research Hospital, Memphis, TN, USA. [11]Duke University Medical Center, Department of Medicine, Durham, NC, USA. [12]Departments of Medicine, Oncology, and Health Policy and Management, Johns Hopkins Schools of Medicine and Public Health, Baltimore, MD, USA. [13]Division of Hematology and Oncology, The Hospital for Sick Children, University of Toronto, Toronto, ON, Canada. ✉e-mail: Aron.onerup@gu.se

**Table 1 | Demographic and lifestyle variables of survivors according to the healthy lifestyle score at baseline**

| | Unhealthy Lifestyle[a] N (%) | Moderately HealthyLifestyle[a] N (%) | Healthy Lifestyle[a] N (%) | Total |
|---|---|---|---|---|
| Number Of Participants | **4782 (25.6)** | **7948 (42.6)** | **5934 (31.8)** | **18664** |
| Mean number of questionnaires (Range) | **4 (2–7)** | **4 (2–7)** | **4 (2–7)** | **4 (2–7)** |
| Sex, Female | 2258 (47.9) | 3928 (49.4) | 2928 (48.9) | 9114 (48.9) |
| Age At Start of Follow-Up, Median (IQR) | 26.2 (22.8–31.1) | 25.3 (21.9–30.2) | 24.4 (21.2–28.9) | 25.2 (21.9–30.0) |
| Years Of Follow-Up from Start of follow-up, Median (IQR) | 12.6 (9.7–23.6) | 13.0 (9.8–24.9) | 13.1 (10.2–25.6) | 12.9 (9.9–24.9) |
| Race/Ethnicity | | | | |
| *Non-Hispanic White* | 3902 (81.1) | 6519 (81.2) | 4876 (81.2) | 15,297 (81.2) |
| *Non-Hispanic Black* | 251 (5.3) | 430 (5.7) | 273 (4.6) | 954 (5.3) |
| *Hispanic* | 344 (7.8) | 574 (8.1) | 414 (7.7) | 1332 (7.9) |
| *Other* | 285 (5.8) | 425 (5.0) | 371 (6.5) | 1081 (5.7) |
| Educational Attainment | | | | |
| *≤High school graduate or GED* | 1754 (36.6) | 2335 (30.3) | 1328 (22.7) | 5417 (29.5) |
| *Some college* | 1852 (39.3) | 2974 (37.9) | 2162 (37.1) | 6988 (38.0) |
| *College graduate or more* | 1145 (24.0) | 2579 (31.8) | 2417 (40.2) | 6141 (32.5) |
| Household Income, USD per year | | | | |
| *<20,000* | 1317 (32.1) | 1668 (24.8) | 1018 (20.2) | 4003 (25.2) |
| *20–39,000* | 741 (16.1) | 1082 (14.6) | 741 (13.0) | 2564 (14.5) |
| *40–79,000* | 1187 (24.5) | 1958 (24.6) | 1464 (25.5) | 4609 (24.8) |
| *≥80,000* | 1365 (27.3) | 2952 (36.0) | 2523 (41.3) | 6840 (35.5) |
| Marital Status | | | | |
| *Married/living as married* | 1999 (41.5) | 2851 (34.3) | 1859 (30.2) | 6709 (34.8) |
| *Separated/divorced/Widowed* | 424 (8.5) | 453 (5.8) | 240 (3.8) | 1117 (5.9) |
| *Never married/lived as married* | 2336 (50.0) | 4618 (59.8) | 3823 (66.0) | 10777 (59.3) |
| Has health Insurance | 3781 (78.8) | 6831 (85.6) | 5304 (89.1) | 15916 (85.0) |
| Smoking History | | | | |
| *Never* | 1337 (29.0) | 5977 (76.3) | 5890 (100.0) | 13,204 (71.9) |
| *Ever* | 3369 (71.0) | 1893 (23.7) | 0 (0.0) | 5262 (28.1) |
| *Past* | 1436 (30.9) | 975 (12.1) | 0 (0.0) | 2411 (13.0) |
| *Current* | 1901 (39.3) | 900 (11.3) | 4 (0.1) | 2805 (14.9) |
| Heavy/Risky Drinking | 1831 (39.7) | 1164 (15.5) | 451 (7.7) | 3446 (19.2) |
| BMI, kg/m$^2$ | | | | |
| *< 18.5 (score = 1)* | 147 (3.0) | 469 (5.6) | 386 (6.1) | 1002 (5.1) |
| *18.5–24.9 (score = 1)* | 1412 (28.8) | 3713 (46.2) | 4189 (70.7) | 9314 (49.6) |
| *25- < 30 (score = 0.5)* | 1193 (25.2) | 2522 (31.8) | 1342 (23.1) | 5057 (27.3) |
| *≥30 (score = 0)* | 2004 (43.0) | 1195 (16.4) | 0 (0.0) | 3199 (18.0) |
| Physical activity, MET-h/wk | | | | |
| *0 (score = 0)* | 3137 (64.6) | 3343 (40.7) | 0 (0.0) | 6480 (33.8) |
| *3–6 (score = 0.5)* | 885 (19.2) | 2095 (26.4) | 1681 (27.6) | 4661 (24.9) |
| *9–12 (score = 1)* | 422 (9.0) | 1510 (19.9) | 2590 (43.8) | 4522 (24.7) |
| *15–21 (score = 1)* | 326 (7.2) | 983 (13.0) | 1640 (28.6) | 2949 (16.5) |

Reported percentages were weighted to account for under-sampling of ALL survivors in the latter recruitment era (1987–1999).

*GED* General Educational Development, *IQR* interquartile range, *MET* metabolic equivalent of task.

[a]Defined by the combined scores for smoking, alcohol consumption, body mass index, and physical activity. The four scores were added to a combined lifestyle score (0–4 points) at each questionnaire and categorized as unhealthy (0–2), moderately healthy (2.5–3), or healthy (3.5–4).

including the absence of smoking and risky alcohol consumption, maintaining a healthy weight, and engaging in physical activity[12–15]. Lifestyle has also been identified as a potential modifiable risk factor in childhood cancer survivors, which has led to the inclusion of general recommendations on healthy lifestyle maintenance in follow-up guidelines[9–11]. Data from the Childhood Cancer Survivor Study (CCSS) shows that exercise and a combined healthy lifestyle score are associated with lower mortality in childhood cancer survivors[6,16], and that survivors of Hodgkin lymphoma with the highest level of vigorous exercise have a lower risk for subsequent major cardiovascular

events[17]. According to a recent systematic review, there was low-quality evidence that a sedentary lifestyle and unfavorable diet are associated with higher risk for the metabolic syndrome (including central obesity, hypertension, dyslipidemia, and/or impaired glucose metabolism), while there was low quality evidence that smoking was not associated with the metabolic syndrome in childhood cancer survivors[18]. In summary, there is limited evidence regarding whether adherence to a healthy lifestyle may prevent treatment-associated CHCs in survivors and no studies on the size of the impact of lifestyle on CHCs or the differential potential of lifestyle interventions in

**Table 2 | Childhood cancer diagnosis and treatment variables of survivors according to the healthy lifestyle score at baseline**

| | Unhealthy Lifestyle[a]<br>N (%) | Moderately Healthy Lifestyle[a]<br>N (%) | Healthy Lifestyle[a]<br>N (%) | Total |
|---|---|---|---|---|
| Number Of Participants | **4782 (25.6)** | **7948 (42.6)** | **5934 (31.8)** | **18664** |
| Age at diagnosis, Median (IQR) | 7.1 (3.5–13.3) | 6.6 (3.1–12.8) | 6.2 (3.1–12.2) | 6.6 (3.2–12.7) |
| *0–4* | 1674 (37.8) | 2994 (40.7) | 2314 (42.3) | 6982 (40.5) |
| *5–9* | 1059 (23.9) | 1785 (23.5) | 1432 (25.0) | 4276 (24.1) |
| *10–14* | 1136 (21.7) | 1774 (20.3) | 1273 (19.2) | 4183 (20.3) |
| *15 +* | 913 (16.6) | 1395 (15.4) | 915 (13.5) | 3223 (15.1) |
| Year Of Primary Cancer Diagnosis | | | | |
| *1970–79* | 1319 (23.8) | 2241 (24.5) | 1553 (22.5) | 5113 (23.7) |
| *1980–89* | 1849 (37.2) | 3166 (37.7) | 2329 (36.5) | 7344 (37.2) |
| *1990–99* | 1614 (39.0) | 2541 (37.7) | 2052 (41.0) | 6207 (39.1) |
| Type Of Primary Cancer Diagnosis | | | | |
| *Leukemia* | 1487 (40.5) | 2422 (39.6) | 1897 (41.4) | 5806 (40.4) |
| *Acute lymphoblastic leukemia* | 1276 (36.7) | 2049 (35.5) | 1579 (36.8) | 4904 (36.2) |
| *Acute myeloid leukemia* | 157 (2.8) | 287 (3.1) | 247 (3.6) | 691 (3.2) |
| *Other leukemia* | 54 (1.0) | 86 (0.9) | 71 (1.0) | 211 (1.0) |
| *Hodgkin lymphoma* | 662 (12.0) | 992 (10.9) | 710 (10.3) | 2364 (11.0) |
| *Non-Hodgkin lymphoma* | 447 (8.1) | 639 (7.0) | 507 (7.4) | 1593 (7.4) |
| *Central nervous system tumors* | 740 (13.4) | 1456 (15.9) | 946 (13.7) | 3142 (14.6) |
| *Wilms tumor* | 419 (7.6) | 677 (7.4) | 578 (8.4) | 1674 (7.8) |
| *Neuroblastoma* | 301 (5.4) | 539 (5.9) | 415 (6.0) | 1255 (5.8) |
| *Soft tissue sarcoma* | 308 (5.6) | 552 (6.0) | 421 (6.1) | 1281 (5.9) |
| *Bone sarcoma* | 418 (7.5) | 671 (7.3) | 460 (6.7) | 1549 (7.2) |
| Treatment exposures | | | | |
| *Radiation Treatment* | 2355 (48.5) | 4268 (53.2) | 3080 (50.7) | 9703 (51.2) |
| *Alkylating Agents* | 2253 (51.8) | 3897 (53.5) | 3016 (54.9) | 9166 (53.5) |
| *Anthracyclines* | 2089 (52.4) | 3487 (52.1) | 2681 (52.3) | 8257 (52.3) |
| *Platinum derivates* | 396 (7.7) | 784 (9.1) | 581 (8.9) | 1761 (8.7) |
| *Epipodophyllotoxins* | 727 (20.1) | 1306 (20.9) | 960 (19.7) | 2993 (20.3) |
| *Intrathecal methotrexate* | 1513 (43.9) | 2405 (41.5) | 1893 (43.3) | 5811 (42.7) |
| *Intravenous methotrexate* | 1061 (29.5) | 1710 (28.2) | 1295 (28.5) | 4066 (28.6) |
| *Corticosteroids* | 1908 (52.0) | 3009 (49.2) | 2439 (52.0) | 7356 (50.8) |
| *Hematopoietic Cell Transplant* | 163 (3.4) | 374 (4.9) | 315 (5.6) | 852 (4.7) |

Reported percentages were weighted to account for under-sampling of ALL survivors in the latter recruitment era (1987–1999).

[a]Defined by the combined scores for smoking, alcohol consumption, body mass index, and physical activity. The four scores were added to a combined lifestyle score (0–4 points) at each questionnaire and categorized as unhealthy (0–2), moderately healthy (2.5–3), or healthy (3.5–4).

survivors and non-survivor controls. Here, we show that an unhealthy lifestyle is associated with higher risk of subsequent hypertension, dyslipidemia, diabetes, heart attack, heart failure, valvular disease, joint replacement, anxiety, depression, and impaired physical and mental quality of life in childhood cancer survivors. The proportion of disease attributable to lifestyle exceeds that of chemotherapy and radiotherapy for hypertension, diabetes, joint replacement, anxiety, depression, and impaired physical and mental quality of life. Furthermore, the potential benefit of a healthy lifestyle is greater in childhood cancer survivors than in sibling comparators for diabetes, heart failure, valvular heart disease, arrhythmia, anxiety, depression, and impaired mental and physical quality of life.

## Results

### Baseline characteristics

Table 1 shows the baseline characteristics of the 18,664 survivors (9,114, 48.9% female) included in this study by lifestyle score, with participant flow charts illustrated in Supplementary Figs. 1 & 2. The median age at the start of follow-up was 25 (interquartile range [IQR]

22–30) years, and participants were followed for a median of 13 (IQR 10–25) years. Across the lifestyle categories, participants were similar in terms of sex, age, race/ethnicity, age at and year of cancer diagnosis, childhood cancer diagnoses and treatment exposures, Table 2. Participants with an unhealthy lifestyle score were less likely to have college education, a high income, and health insurance, and more likely to be married than those with a healthy lifestyle. The number of events by lifestyle score category is shown in Supplementary Table 1. All results below are summarized in Fig. 1.

### Associations between lifestyle score and CHCs

Figures 2 and 3 show the rate ratio (RR) and odds ratio (OR) for the lifestyle score and CHCs. Unhealthy lifestyle was associated with statistically significantly higher rates of subsequent hypertension (RR 1.5), dyslipidemia (RR 1.3), diabetes (RR 2.8), heart attack (RR 1.4), heart failure (RR 1.4), heart valve disease (RR 1.8), joint replacement (RR 1.7), anxiety (OR 1.8), depression (OR 1.6), and impaired physical (OR 2.1) and mental (OR 1.8) quality of life (QoL), compared to healthy lifestyle. The lifestyle score was not

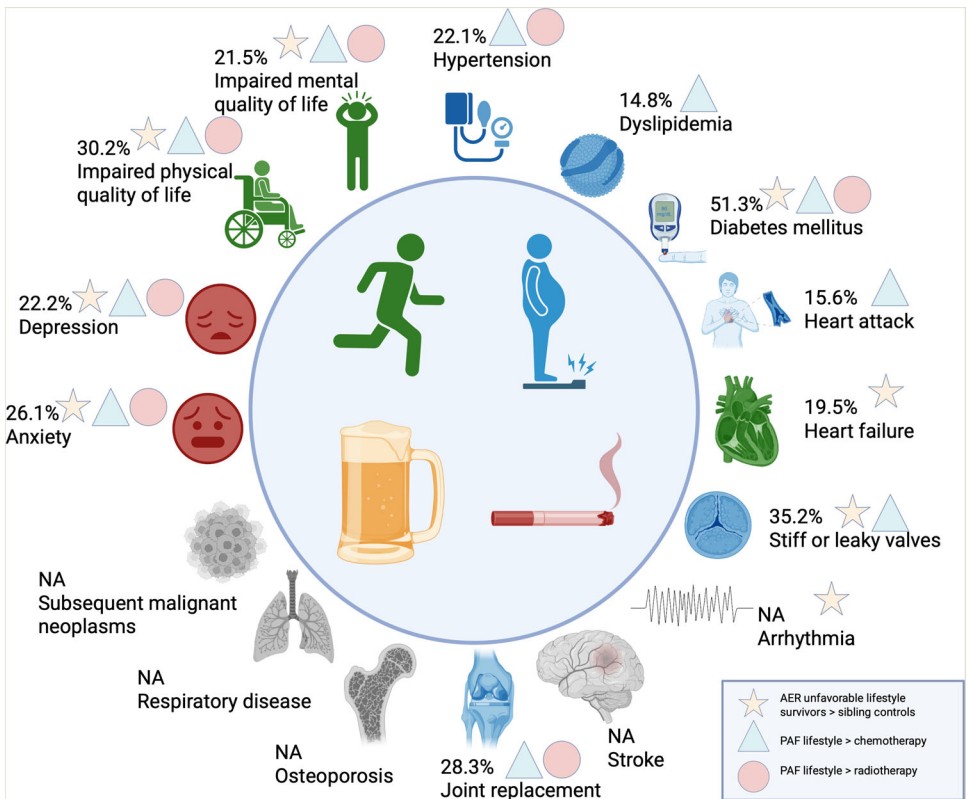

**Fig. 1 | The central illustration summarizes the population attributable fractions (PAF) of each chronic health condition attributed to unfavorable lifestyle score among participants in the Childhood Cancer Survivor Study,** *n* = 18,664. Coloring of symbols indicate the individual lifestyle variables most strongly contributing to the results, with lack of exercise in green, overweight/obesity in blue, smoking in red and heavy/risky drinking in yellow. Chronic health conditions not associated with the lifestyle score are not colored and the estimation of PAF is not applicable (NA). The figure also indicates for which chronic health conditions the PAF of lifestyle score exceeds those of chemotherapy (triangle) or radiotherapy (circle). Stars indicate that the absolute excess risk of not having a healthy lifestyle score is significantly higher in childhood cancer survivors than in sibling controls. Created in BioRender. Onerup, A. (2026) https://BioRender.com/jge2wr2.

associated with subsequent stroke, osteoporosis, subsequent malignant neoplasms (SMN), arrhythmia, or respiratory disease. The only interactions between the lifestyle score and attained age were seen for associations with hypertension (stronger associations in the 30 s) and SMN (associations seen only in the 20 s), Supplementary Tables 2 and 3.

### Association between individual lifestyle factors and CHCs
The associations between lifestyle and CHC were explained mainly by associations between overweight or obese body mass index (BMI) and/ or low physical activity (Supplementary Tables 4 and 5). All outcomes associated with the combined lifestyle score had associations with at least one of the two. Additionally, BMI and/or physical activity were significantly associated with several outcomes that were not associated with the combined lifestyle score, including arrhythmia (BMI), respiratory disease (BMI and physical activity), and SMN (physical activity). Heavy/risky drinking was only associated with stroke and anxiety, while smoking was associated with higher risk of stroke, osteoporosis, SMN, anxiety, depression, and impaired physical and mental QoL.

### Population attributable fractions for lifestyle score
The PAF of lifestyle exceeded those of both chemotherapy and radiotherapy for hypertension (22%), diabetes (51%), joint replacement (28%), anxiety (26%), depression (22%), and impaired physical (30%) and mental (22%) QoL (Fig. 4, Supplementary Table 6). They also exceeded those of chemotherapy for dyslipidemia (15%), heart attack (16%), valvular disease (35%), and were considerable for heart failure (20%).

### Population attributable fractions for lifestyle factors
The PAFs of lifestyle were explained mainly by unfavorable BMI and low physical activity, since these were both more prevalent and had stronger associations with the outcomes than smoking and risky drinking (Fig. 5, Supplementary Table 7). The largest contributions came from overweight/obesity for hypertension (26%), dyslipidemia (19%), diabetes (59%), heart attack (14%), valvular heart disease (25%), arrhythmia (9%), and joint replacement (26%). Physical activity could be attributed more than the other lifestyle variables for heart failure (14%), respiratory disease (16%), SMN (7%), and impaired physical (20%) and mental (11%) QoL. Of the four lifestyle factors, smoking was most attributable for stroke (7%), osteoporosis (7%), anxiety (16%), and depression (10%).

### Absolute excess risk for lifestyle score
Figures 6, 7, and 8 show the absolute excess risks (AERs) associated with an unhealthy or moderately healthy lifestyle score, compared to a healthy lifestyle score, and the comparison between childhood cancer survivors and sibling controls (baseline demographics shown in Supplementary Table 8). For diabetes, heart failure, valvular heart disease, arrhythmia, anxiety, depression, and impaired mental and physical QoL, the AERs for moderately healthy/unhealthy lifestyle score were significantly higher in childhood cancer survivors than in sibling controls, Supplementary Table 9. Conversely, the AER for moderately healthy lifestyle was higher in siblings than in survivors for heart attack, while the directions were mixed for moderately healthy and unhealthy lifestyle for hypertension, stroke, and osteoporosis.

| Chronic health conditions | Lifestyle score | RR | 95% CI | P value |
|---|---|---|---|---|
| Hypertension | Moderately healthy | 1.3 | 1.2 – 1.5 | <.001 |
| | Unhealthy | 1.5 | 1.3 – 1.7 | <.001 |
| Dyslipidemia | Moderately healthy | 1.2 | 1.0 – 1.3 | 0.011 |
| | Unhealthy | 1.3 | 1.1 – 1.5 | <.001 |
| Diabetes mellitus | Moderately healthy | 2.1 | 1.6 – 2.6 | <.001 |
| | Unhealthy | 2.8 | 2.2 – 3.7 | <.001 |
| Heart attack | Moderately healthy | 1.1 | 0.9 – 1.5 | 0.29 |
| | Unhealthy | 1.4 | 1.1 – 1.7 | 0.015 |
| Heart failure | Moderately healthy | 1.3 | 1.0 – 1.6 | 0.071 |
| | Unhealthy | 1.4 | 1.1 – 1.8 | 0.019 |
| Stiff or leaky valves | Moderately healthy | 1.7 | 1.2 – 2.4 | 0.007 |
| | Unhealthy | 1.8 | 1.2 – 2.7 | 0.006 |
| Arrhythmia | Moderately healthy | 1.3 | 1.0 – 1.7 | 0.055 |
| | Unhealthy | 1.3 | 1.0 – 1.7 | 0.064 |
| Stroke | Moderately healthy | 1.2 | 0.9 – 1.5 | 0.29 |
| | Unhealthy | 1.1 | 0.8 – 1.4 | 0.7 |
| Joint replacement | Moderately healthy | 1.3 | 0.9 – 2.1 | 0.18 |
| | Unhealthy | 1.7 | 1.1 – 2.7 | 0.013 |
| Osteoporosis | Moderately healthy | 1.2 | 0.9 – 1.5 | 0.19 |
| | Unhealthy | 1.1 | 0.8 – 1.4 | 0.7 |
| Respiratory disease | Moderately healthy | 1.2 | 0.9 – 1.5 | 0.32 |
| | Unhealthy | 1.3 | 1.0 – 1.8 | 0.061 |
| Subsequent malignant neoplasms | Moderately healthy | 1.1 | 1.0 – 1.2 | 0.18 |
| | Unhealthy | 1.0 | 0.8 – 1.1 | 0.58 |

0.5    1    2    4

**Fig. 2 | Rate ratios for unhealthy and moderately healthy lifestyle scores, in reference to healthy lifestyle score, for chronic health conditions.** Bars represent rate ratios with 95% confidence intervals. Piecewise exponential models were used. All analyses were adjusted for attained age as cubic splines, sex, race/ethnicity, age at cancer diagnosis, socioeconomic status (including education, marriage status, household income, and insurance), and treatment variables chosen from backward selection for each outcome. All statistical tests were two-sided. No correction was performed for multiple testing since all outcomes were distinct hypotheses of individual interest. Number of participants in the total population is 18,664, with analysis of each outcome differing slightly due to pre-existing conditions, as reported in Supplementary Table 1. Source data are provided as a Source Data file. CI = confidence interval. RR = rate ratio.

| Emotional distress and quality of life | Lifestyle score | OR | 95% CI | P value |
|---|---|---|---|---|
| Anxiety | Moderately healthy | 1.3 | 1.1 – 1.6 | 0.002 |
| | Unhealthy | 1.8 | 1.5 – 2.2 | <.001 |
| Depression | Moderately healthy | 1.3 | 1.1 – 1.5 | <.001 |
| | Unhealthy | 1.6 | 1.4 – 1.9 | <.001 |
| Impaired physical QoL | Moderately healthy | 1.5 | 1.3 – 1.7 | <.001 |
| | Unhealthy | 2.1 | 1.8 – 2.4 | <.001 |
| Impaired mental QoL | Moderately healthy | 1.3 | 1.2 – 1.4 | <.001 |
| | Unhealthy | 1.8 | 1.6 – 2.0 | <.001 |

0.5    1    2    4

**Fig. 3 | Odds ratios for unhealthy and moderately healthy lifestyle scores, in reference to healthy lifestyle score, for emotional distress and quality of life.** Bars represent odds ratios with 95% confidence intervals, using logistic regressions. Results are from longitudinal data with generalized estimating equations. All analyses were adjusted for attained age as cubic splines, sex, race/ethnicity, age at cancer diagnosis, socioeconomic status (including education, marriage status, household income, and insurance), and treatment variables chosen from backward selection for each outcome. All statistical tests were two-sided. No correction was performed for multiple testing since all outcomes were distinct hypotheses of individual interest. Number of participants in the total population is 18,664, with analysis of each outcome differing slightly due to pre-existing conditions, as reported in Supplementary Table 1. Source data are provided as a Source Data file. CI = confidence interval. OR = odds ratio.

## Discussion

In this observational cohort study of >18,000 childhood cancer survivors followed longitudinally over 30 years, we demonstrated that despite four decades of knowledge that primary cancer treatment exposures drive the high risk for early onset of life-altering chronic health conditions, lifestyle was associated with several CHCs, consistent with similar observations regarding the relationship between lifestyle and health in the general population. Furthermore, we observed that the population attributable fraction of lifestyle was comparable to, or greater than, those of radiation and chemotherapy exposure for several CHCs and for health-related QoL. Finally, we identified that healthy lifestyle habits are associated with greater benefit in childhood cancer survivors than in the general population for reducing the risk of developing heart failure, valvular heart disease, emotional distress, and QoL impairments.

Given the well-established role of radiation and chemotherapy with early onset of chronic health conditions in survivors of childhood cancer[5], the association between lifestyle and CHC was uncertain. However, given the vast literature on the associations between lifestyle and CHCs in the general population[12–14], it is reassuring to see the significant associations between lifestyle and CHCs in childhood survivors in the setting of exposure to cancer therapies[12,19,20]. Since interventions to improve lifestyle are resource intensive[21], it was necessary to establish this relationship in childhood cancer survivors before endorsing the need to develop such interventions. This shows promise for a modifiable risk factor that can be intervened upon both early after therapy and perhaps years and decades after the childhood cancer treatment to mitigate its detrimental effects. Furthermore, with chronic health conditions considered a phenotype of accelerated aging in childhood cancer survivors[22], it is possible that these result

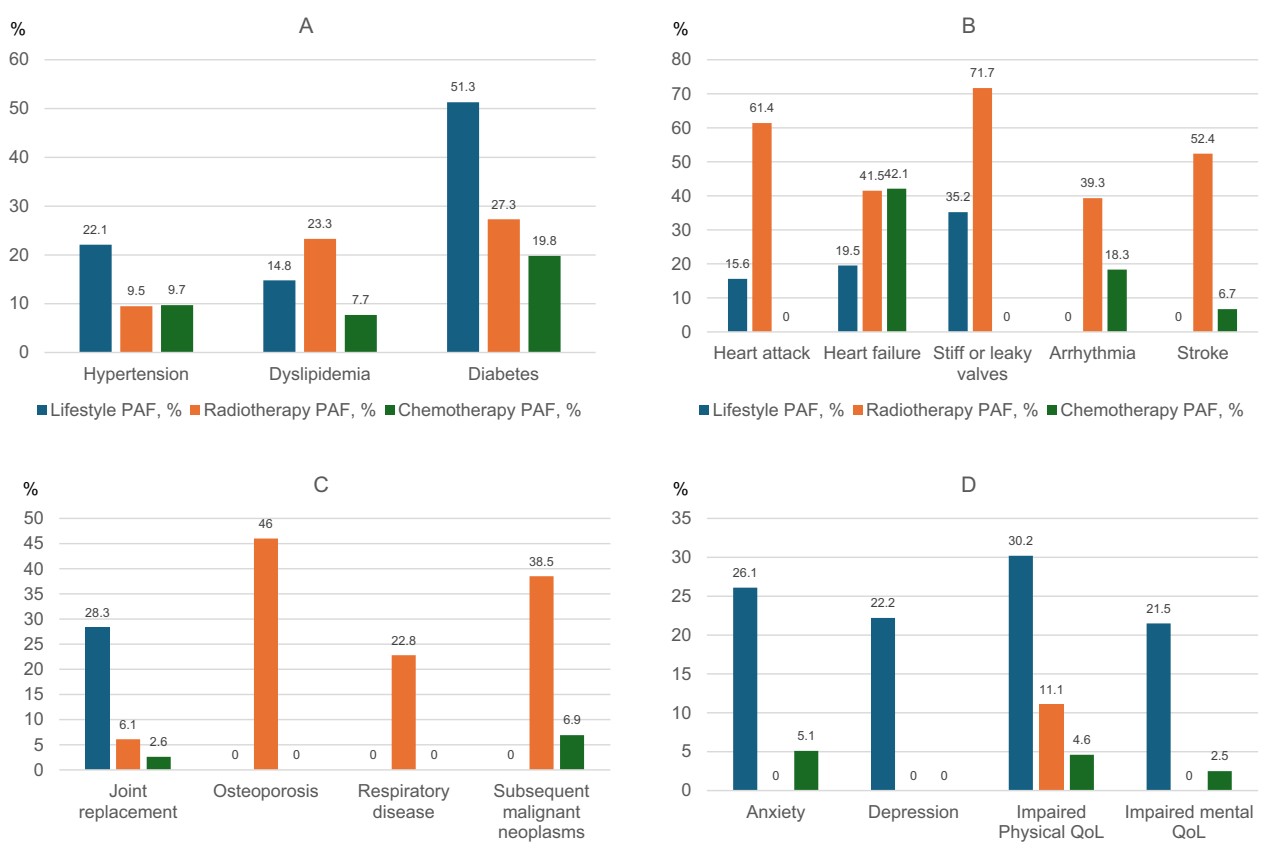

**Fig. 4 | Population attributable fractions for unfavorable lifestyle, radiotherapy, and chemotherapy for chronic health conditions.** Bar chart illustrating the population attributable fractions (PAF) for unfavorable lifestyle, radiotherapy, and chemotherapy for **A** cardiometabolic disease, **B** cardiovascular disease, **C** other, and **D** emotional distress and impaired quality of life (QoL). Number of participants in the total population is 18,664, with analysis of each outcome differing slightly due to pre-existing conditions, as reported in Supplementary Table 1. Source data are provided as a Source Data file.

signal potential reversal of this accelerated aging by healthy lifestyle, as has been suggested in cross-sectional studies[23]. This potential for mitigation of treatment-related accelerated aging, combined with the higher AERs in survivors with unhealthy lifestyles, show potential for lifestyle interventions as part of precision health initiatives in childhood cancer survivors.

Our results now substantiate previous reports that have suggested associations between a healthy lifestyle and reduced risk for excess mortality[6,16], physical activity and cardiovascular events in survivors of Hodgkin lymphoma[17], and the low quality evidence on associations between physical activity and components of the metabolic syndrome in childhood cancer survivors[18]. To our knowledge, there are no previous studies that have reported associations between physical activity and outcomes other than cardiovascular risk factors, and there are no previous reports of associations between e.g. smoking or alcohol consumption and any of these risk factors in childhood cancer survivors. Furthermore, no previous study has compared the potential of combined lifestyle on CHCs in adult childhood cancer survivors with the impacts of chemotherapy and radiotherapy exposures or compared the benefit of healthy habits on the risk of CHCs in childhood cancer survivors and non-survivor controls. Our results suggest that if we want to identify childhood cancer survivors at high risk for CHCs, we should include lifestyle in our risk-based assessments, alongside treatment exposures.

Since lifestyle has previously been identified as one of few potential modifiable risk factors for CHCs in childhood cancer survivors[9,18], there are general recommendations on the benefit of a healthy lifestyle in national and international long-term follow-up guidelines[9–11]. General recommendations on lifestyle behavior change have shown little or no effect and are generally used as the control condition in lifestyle intervention trials[24,25]. Our results suggest that the same behavior change will reduce the risk for diabetes, heart failure, valvular heart disease, arrhythmia, anxiety, depression, and QoL impairments more in childhood cancer survivors than in the general population. Unfortunately, validated lifestyle interventions may not be effective in childhood cancer survivors despite being effective in the general population[26]. Hence, effective lifestyle interventions aimed at long-term change need to be further developed and tested in childhood cancer survivors. As lifestyle interventions may only achieve a partial shift towards healthier behaviors, it is promising that the AERs were larger in survivors than sibling controls even for moderately unhealthy lifestyle. Thus, interventions that successfully shift survivors partially towards a healthier lifestyle may be beneficial.

While efforts should be made to reduce the prevalence of smoking and risky drinking, our results indicate that promoting physical activity and/or attaining a healthy weight may be most beneficial in reducing CHCs in survivors. This is also in line with previous studies reporting that childhood cancer survivors have lower prevalence of smoking and risky drinking but higher prevalence of inadequate physical activity and obesity than the general population[27–30]. Since physical activity and diet are more effective at preventing than treating obesity in the general population[31], it has been proposed that interventions should start early in childhood cancer survivors[32]. However, whether this can be done in an effective way needs to be shown[32,33].

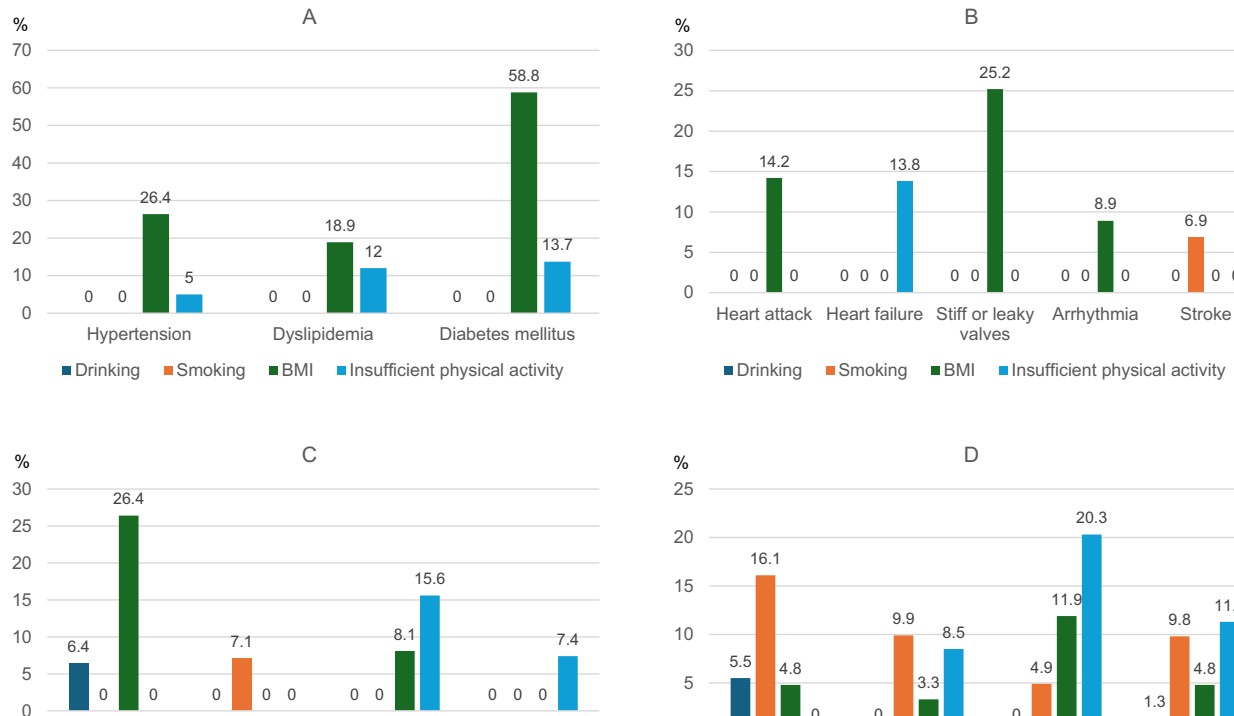

**Fig. 5 | Population attributable fractions for individual lifestyle factors and chronic health conditions.** Bar chart illustrating the population attributable fractions (PAF) for heavy/risky drinking, smoking, unfavorable body mass index (BMI), and insufficient physical activity for (A) cardiometabolic disease, (B) cardiovascular disease, (C) other, and (D) emotional distress and impaired quality of life (QoL). Number of participants in the total population is 18,664, with analysis of each outcome differing slightly due to pre-existing conditions, as reported in Supplementary Table 1. Source data are provided as a Source Data file.

Optimal intervention timing requires balancing the difficulty of establishing long-term lifestyle behaviors during the dynamic phases of active treatment while intervening before the occurrence of lifestyle-associated morbidity.

The large sample size and long follow-up are strengths of this study. This design enabled longitudinal follow-up, removing the risk of reverse causality seen in cross-sectional studies of lifestyle and CHCs[34,35]. The multiple assessments allowed for analyses of the dynamic exposure to lifestyle behaviors rather than a single assessment. The inclusion of a population of adult survivors treated for a variety of childhood cancers allows for generalizing the results to other general childhood cancer survivor populations[17]. However, while the rate ratios and absolute excess risks should be generalizable to other populations, the PAFs in our study are not generalizable to other populations with different distributions of, for example, obesity[36].

In this study, we included BMI mainly to reflect unhealthy diet. However, BMI reflects both diet and physical activity patterns, making it hard to differentiate the specific contributions from either of the two. As the main purpose of this study is to assess the contributions from lifestyle to CHCs, this differentiation is not central to our results. However, the lack of dietary data means that our study provides no information on associations between specific types of diet and CHCs. Furthermore, obesity is also a result of genetic predisposition. However, with obesity in adults 18-25 increasing by six-fold in the US between 1976 and 2018[37], lifestyle is key to the development and prevention of obesity[38]. While unfavorable BMI may also be related to childhood cancer treatment exposures, our analyses were adjusted for treatment exposures so the observed BMI association should not be confounded by treatment exposure.

Both the exposures and outcomes in our study were self-reported, reducing sensitivity. However, we do not consider it plausible that self-reporting of chronic health conditions is dependent on lifestyle or vice versa. Hence, our study might have yielded lower rate ratios and PAFs than had the exposures or outcomes been clinically ascertained. The use of siblings as a reference group for the comparison of AER is a limitation. Siblings may not represent the general population in some respects, as they often share many of the same social, biological, and environmental exposures as the survivors. However, using siblings as the reference group should yield conservative estimates of the absolute excess risk. Furthermore, the shared uncontrolled confounders may also be considered a strength[39]. As we considered all CHCs to represent individual hypotheses, we did not adjust for multiple comparisons. This may increase risk for Type 1 error and should be taken into account when interpreting our results. In the analyses of emotional distress and health-related QoL, we used generalized estimating equations (GEE) to account for within-individual correlation across repeated measurements. While GEE provides robust population-averaged estimates, it assumes that data are missing completely at random and does not provide a likelihood function, limiting formal model comparison.

We are not aware of any other study that will have the statistical power to confirm our results within foreseeable time. While causality cannot be claimed in an observational study, it should be noted that longitudinal observational studies such as ours with appropriate temporal ordering, large sample sizes, and careful control of confounding can contribute meaningfully to causal inference, particularly when supported by biological plausibility, consistency with prior evidence from other populations, and dose-response patterns. These results should rather be considered evidence of the potential of

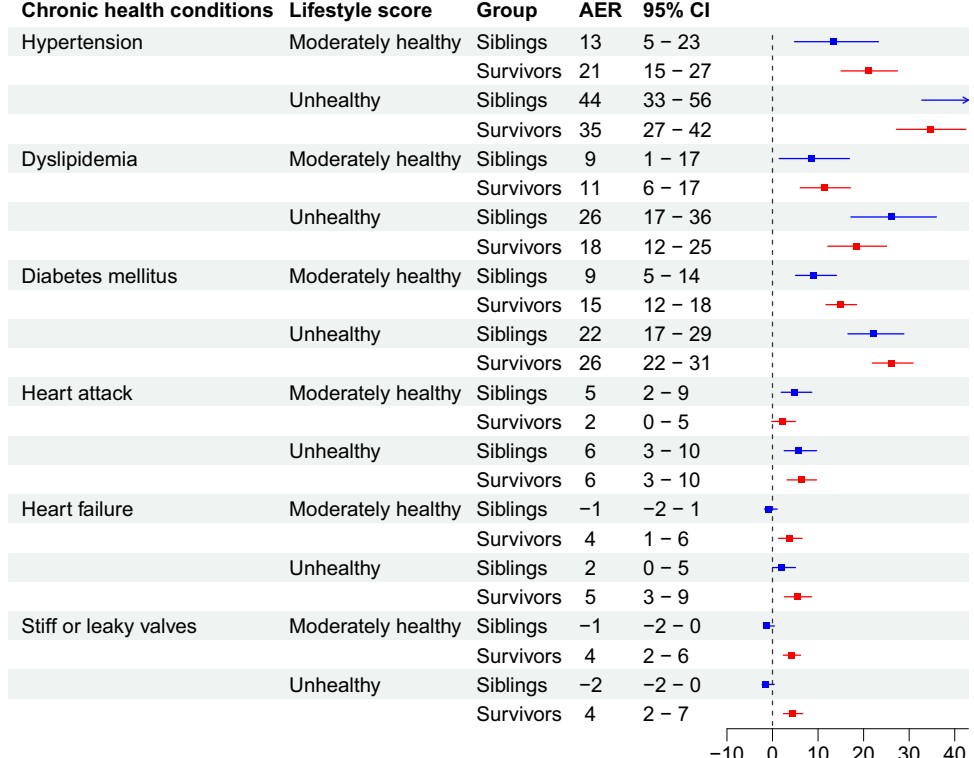

| Chronic health conditions | Lifestyle score | Group | AER | 95% CI |
|---|---|---|---|---|
| Hypertension | Moderately healthy | Siblings | 13 | 5 – 23 |
| | | Survivors | 21 | 15 – 27 |
| | Unhealthy | Siblings | 44 | 33 – 56 |
| | | Survivors | 35 | 27 – 42 |
| Dyslipidemia | Moderately healthy | Siblings | 9 | 1 – 17 |
| | | Survivors | 11 | 6 – 17 |
| | Unhealthy | Siblings | 26 | 17 – 36 |
| | | Survivors | 18 | 12 – 25 |
| Diabetes mellitus | Moderately healthy | Siblings | 9 | 5 – 14 |
| | | Survivors | 15 | 12 – 18 |
| | Unhealthy | Siblings | 22 | 17 – 29 |
| | | Survivors | 26 | 22 – 31 |
| Heart attack | Moderately healthy | Siblings | 5 | 2 – 9 |
| | | Survivors | 2 | 0 – 5 |
| | Unhealthy | Siblings | 6 | 3 – 10 |
| | | Survivors | 6 | 3 – 10 |
| Heart failure | Moderately healthy | Siblings | −1 | −2 – 1 |
| | | Survivors | 4 | 1 – 6 |
| | Unhealthy | Siblings | 2 | 0 – 5 |
| | | Survivors | 5 | 3 – 9 |
| Stiff or leaky valves | Moderately healthy | Siblings | −1 | −2 – 0 |
| | | Survivors | 4 | 2 – 6 |
| | Unhealthy | Siblings | −2 | −2 – 0 |
| | | Survivors | 4 | 2 – 7 |

**Fig. 6 | Absolute excess risk per 10,000 person-years for developing chronic health conditions associated with unhealthy or moderately healthy lifestyle score compared to healthy lifestyle in childhood cancer survivors and sibling controls.** Bars represent AER with 95% confidence intervals. Analyses are adjusted for attained age as cubic splines, sex, race/ethnicity, age at cancer diagnosis, socioeconomic status (including education, marriage status, household income, and insurance), and treatment variables chosen from backward selection for each outcome. Number of participants in the total population is 18,664, with analysis of each outcome differing slightly due to pre-existing conditions, as reported in Supplementary Table 1. Source data are provided as a Source Data file. AER = absolute excess risk. CI = confidence interval.

lifestyle to lessen CHCs in childhood cancer survivors and motivate efforts in determining effects from lifestyle in intervention trials.

In this work we show that a significant proportion of chronic health conditions in childhood cancer survivors can be attributed to lifestyle. Unlike previous cancer treatment exposures, lifestyle can be modified. Action is needed to further develop and implement effective interventions to improve lifestyle in childhood cancer survivors.

## Methods
### Population
This longitudinal observational study was performed in the CCSS, a multi-institutional, hospital-based retrospectively-constructed cohort with longitudinal follow-up of childhood cancer survivors that were diagnosed and treated at 31 institutions in the US and Canada[4,6,16,17,40,41]. The CCSS included those who had received a cancer diagnosis at age <21 years between 1970 and 1999, and who were alive five years after the diagnosis. The study methods and design have been described in detail previously[42]. This study complies with all relevant ethical regulations. The CCSS was approved by the institutional review boards at each of the 31 participating centers, with current continuing review at St. Jude Children's Research Hospital of January 2026, with expiry date January 30th, 2027. All participants provided written informed consent. Participants received USD 10 per completed follow-up survey. CCSS participants were included in our analysis if they were ≥18 years at their first questionnaire-based assessment and had participated in at least two questionnaires, contributing data on exposures and outcomes from different questionnaires. To compare the associations between lifestyle and CHCs in childhood cancer survivors with the associations in non-survivor controls, we included sibling controls from the CCSS.

### Exposures
Self-reported data on smoking status, alcohol consumption, body weight and height, and vigorous physical activity were collected from each questionnaire. The questions used have been described previously[16,27,28]. Responses for each lifestyle behavior were categorized and scored[6]. Smoking was dichotomized as having ever smoked ≥100 cigarettes (0 point)[6] or not (1 point). Alcohol consumption was dichotomized as having heavy or risky drinking ( > 7 drinks per week or >3 drinks per day for women, and >14 drinks per week or >4 drinks per day for men, 0 point) or not (1 point)[43]. BMI (kg/m$^2$) was calculated using weight and height and categorized as underweight or normal weight ( < 25 kg/m$^2$, 1 point), overweight (25–29.9 kg/m$^2$, 0.5 point), or obese ( ≥ 30 kg/m$^2$, 0 point). Self-reported vigorous physical activity was converted into average metabolic equivalent task (MET) hours per week[16], and was categorized as sedentary (0- < 3 MET-h/wk, 0 point), low physical activity (3–8 MET-h/wk, 0.5 point), or sufficient ( ≥ 9 MET-h/wk, 1 point), approximating to <20, 20 to <60, or ≥60 minutes of vigorous physical activity per week, respectively[6]. The four scores were summed for a combined lifestyle score (0–4 points) at each questionnaire and categorized as unhealthy (0–2), moderately healthy (2.5–3), or healthy (3.5–4). This score has been used previously and was associated with late excess mortality, except our study added 0.5 for overweight, resulting in 0.5 point increase in the score range for our study[6].

### Outcomes
The time of first occurrence of each CHC over time was assessed by a series of questions in every study questionnaire, available at www.ccss.stjude.org. [42,44] Using a well-established algorithm, a multidisciplinary

| Chronic health conditions | Lifestyle score | Group | AER | 95% CI |
|---|---|---|---|---|
| Arrhythmia | Moderately healthy | Siblings | 1 | −2 – 4 |
| | | Survivors | 4 | 1 – 7 |
| | Unhealthy | Siblings | 3 | 0 – 7 |
| | | Survivors | 4 | 1 – 7 |
| Stroke | Moderately healthy | Siblings | 1 | 0 – 4 |
| | | Survivors | 2 | 0 – 5 |
| | Unhealthy | Siblings | 3 | 1 – 6 |
| | | Survivors | 1 | −2 – 4 |
| Joint replacement | Moderately healthy | Siblings | 3 | 1 – 7 |
| | | Survivors | 2 | 0 – 3 |
| | Unhealthy | Siblings | 5 | 2 – 8 |
| | | Survivors | 4 | 2 – 6 |
| Osteoporosis | Moderately healthy | Siblings | 1 | 0 – 4 |
| | | Survivors | 3 | 1 – 6 |
| | Unhealthy | Siblings | 3 | 1 – 7 |
| | | Survivors | 1 | −2 – 4 |
| Respiratory disease | Moderately healthy | Siblings | 2 | −1 – 6 |
| | | Survivors | 2 | 0 – 4 |
| | Unhealthy | Siblings | 4 | 1 – 8 |
| | | Survivors | 4 | 1 – 7 |
| Subsequent malignant neoplasms | Moderately healthy | Siblings | −2 | −6 – 3 |
| | | Survivors | 6 | 1 – 12 |
| | Unhealthy | Siblings | 3 | −2 – 8 |
| | | Survivors | −3 | −8 – 3 |

−10  0  10  20  30  40

**Fig. 7 | Absolute excess risk per 10,000 person-years for developing chronic health conditions associated with unhealthy or moderately healthy lifestyle score compared to healthy lifestyle in childhood cancer survivors and sibling controls.** Bars represent AER with 95% confidence intervals. Analyses are adjusted for attained age as cubic splines, sex, race/ethnicity, age at cancer diagnosis, socioeconomic status (including education, marriage status, household income, and insurance), and treatment variables chosen from backward selection for each outcome. Number of participants in the total population is 18,664, with analysis of each outcome differing slightly due to pre-existing conditions, as reported in Supplementary Table 1. Source data are provided as a Source Data file. AER = absolute excess risk. CI = confidence interval.

| Emotional distress and quality of life | Lifestyle score | Group | Risk reduction | 95% CI |
|---|---|---|---|---|
| Anxiety | Moderately healthy | Siblings | 1.1 | 0.8 – 1.6 |
| | | Survivors | 1.1 | 0.9 – 1.4 |
| | Unhealthy | Siblings | 1.4 | 0.9 – 2.1 |
| | | Survivors | 2.8 | 2.1 – 3.5 |
| Depression | Moderately healthy | Siblings | 1.1 | 0.8 – 1.4 |
| | | Survivors | 1.4 | 1.2 – 1.7 |
| | Unhealthy | Siblings | 1.8 | 1.3 – 2.4 |
| | | Survivors | 3.4 | 2.7 – 4.1 |
| Impaired mental QoL | Moderately healthy | Siblings | 2.5 | 2.1 – 2.8 |
| | | Survivors | 3.1 | 2.8 – 3.4 |
| | Unhealthy | Siblings | 5.6 | 4.5 – 6.7 |
| | | Survivors | 8.5 | 7.6 – 9.3 |
| Impaired physical QoL | Moderately healthy | Siblings | 3.1 | 2.2 – 4.1 |
| | | Survivors | 4.0 | 3.4 – 4.6 |
| | Unhealthy | Siblings | 7.1 | 5.1 – 9.5 |
| | | Survivors | 8.9 | 7.7 – 10.0 |

0  2  4  6  8  10

**Fig. 8 | Risk reduction (probability difference, per 100) for emotional distress and impaired quality of life associated with unhealthy or moderately healthy lifestyle score compared to healthy lifestyle in childhood cancer survivors and sibling controls.** Bars represent Risk reduction with 95% confidence intervals. Analyses were adjusted for attained age as cubic splines, sex, and race/ethnicity. Number of participants in the total population is 18,664, with analysis of each outcome differing slightly due to pre-existing conditions, as reported in Supplementary Table 1. Source data are provided as a Source Data file. CI = confidence interval.

team reviewed and adjudicated all reported conditions, which were graded and scored according to the National Cancer Institute's Common Terminology Criteria for Adverse Events (CTCAE v4.03)[45].

Outcomes of emotional distress included depression and anxiety and were assessed cross-sectionally with the 18-item Brief Symptom Inventory-18 (BSI-18), which includes symptoms over the previous 7 days[46]. Raw scores were converted to T-scores based on U.S. population norms and dichotomized using a cut-point of 63; for depression and anxiety, participants with a T-score ≥63 (90th percentile) were classified as having clinically significant concerns[46]. The Medical Outcomes Short Form-36 (SF-36) was used to evaluate health-related QoL cross-sectionally[46]. Participants answered 36 questions about general health, well-being, and quality of life over the previous 4 weeks. The SF-36 has two component summary scales, physical and

mental (PCS and MCS, respectively). For each summary scale, T scores ≤40, corresponding to 1 standard deviation below the mean of the US general population, were categorized as impaired. The PCS and MCS are referred to as physical and mental QoL in the manuscript.

The following CHCs were prespecified for inclusion in our analysis, all known to be associated with lifestyle in the general population[12]: hypertension (CTCAE grades 2-5), dyslipidemia (CTCAE 2-5), diabetes mellitus (CTCAE 2-5), heart attack (CTCAE 3-5), heart failure (CTCAE 3-5), valvular heart disease (CTCAE 4-5), arrhythmia (CTCAE 2-5), stroke (CTCAE 4-5), joint replacement (CTCAE 3-5), osteoporosis (CTCAE 2-5), respiratory disease (CTCAE 1-5), SMNs (CTCAE 3-5), anxiety, depression, and impaired QoL (PCS and MCS).

### Cancer treatments

Data regarding childhood cancer diagnosis and treatment in the five years following diagnosis, including chemotherapy and radiotherapy exposures, were extracted from medical records of all participants[42]. We included exposures (yes/no) to a range of chemotherapy classes and radiation exposure to relevant organs (yes/no). The treatment exposures to be included in the analyses for each CHC were based international guidelines, where applicable[9,10], and using backward selection for CHCs where no consensus was available. The treatment exposures of interest are listed in Table 1.

### Statistics and reproducibility

Analyses were performed according to a pre-specified statistical analysis plan, available online (https://ccss.stjude.org/design-a-study/process/approved-concept-proposals.html)[47]. No formal sample size/power calculation was performed as this was an analysis of an existing database and there were several primary outcomes[48]. All eligible participants in the CCSS who met the inclusion criteria were included in the analyses. Participants with pre-existing conditions were excluded from specific outcome analyses. No randomization or blinding was performed, as this was an observational study.

Associations between lifestyle score and subsequent CHCs were analyzed with piecewise exponential models with the start of the at-risk time at the first questionnaire (baseline) with information on lifestyle (the exposure) and the end at the earliest of the target CHC(s), death, or last follow-up. Since lifestyle behaviors are dynamic and assessed at multiple questionnaire time-points, they were included as time-varying covariates (changes at the midpoints between adjacent questionnaires) so that changes over time are accounted for in the analyses. Since lifestyle impacts health over long time periods and to minimize the possibility of bias due to reverse causality from lifestyle change or weight loss prompted by chronic illness (such as low physical activity due to progressive heart failure), only lifestyle scores at least five years prior to the event of interest were considered (time-dependent)[6]. Associations between the lifestyle score and emotional distress (depression/anxiety) and health-related QoL were analyzed using logistic regression models with generalized estimating equations to account for potential correlation across longitudinal measurements of the same individual. All analyses were adjusted for potential confounders, selected based on a hypothesized directed acyclic graph: attained age as cubic splines, self-reported sex, race/ethnicity, age at cancer diagnosis, time-dependent socioeconomic status (SES) including education, marriage status, household income, and insurance, and treatment exposures. Socioeconomic status was categorized as in Table 1. The treatment exposures were chosen from backward selection for each outcome. Attained age was modeled using restricted cubic splines with five knots placed at the 5th, 25th, 50th, 75th, and 95th percentiles of the age distribution to allow for non-linear associations with each outcome. To evaluate whether associations between lifestyle and outcomes differed across age, interaction terms between attained age and the lifestyle score were assessed using likelihood ratio tests. Associations between the individual lifestyle components and the outcomes were assessed with piecewise exponential models, as in the primary analyses but including smoking, heavy/risky drinking, physical activity, and BMI independently instead of the lifestyle score. All statistical tests were two-sided. All analyses were performed in SAS 9.4.

The outcomes were analyzed separately, each as an outcome of interest, with death due to other causes in the analysis of a specific CHC being the only competing risk event. No correction was performed for multiple testing since all outcomes were distinct hypotheses of individual interest, and the significance level was set to 5%. Results are presented as RR estimates with standard large-sample 95% confidence intervals (CI). PAF was estimated as (observed event counts–expected event count)/observed, where expected is the count assuming each participant's lifestyle as being healthy. AER was estimated (observed event counts–expected event count)/person-year in childhood cancer survivor and in sibling controls. For emotional distress (depression/anxiety) and QoL, risk reduction was estimated: (expected probability in the unhealthy–expected probability assuming this group is healthy). Nonparametric bootstrap was used to test statistical significance of the differences in AERs by resampling families[49].

### Reporting summary

Further information on research design is available in the Nature Portfolio Reporting Summary linked to this article.

### Data availability

The Childhood Cancer Survivor Study (CCSS) is a US National Cancer Institute funded resource (U24 CA55727) to promote and facilitate research among long-term survivors of cancer diagnosed during childhood and adolescence. CCSS data are publicly available on dbGaP at https://www.ncbi.nlm.nih.gov/gap/. through its accession number phs001327.v2.p1. and on the St Jude Survivorship Portal within the St. Jude Cloud at https://www.stjude.cloud/research-domains/cancer-survivorship. In addition, utilization of the CCSS data that leverages the expertise of CCSS Statistical and Survivorship research and resources will be considered on a case-by case basis. For this utilization, a research Application Of Intent followed by an Analysis Concept Proposal must be submitted for evaluation by the CCSS Publications Committee. Users interested in utilizing this resource are encouraged to visit http://ccss.stjude.org. Full analytical data sets associated with CCSS publications since January of 2023 are also available on the St. Jude Survivorship Portal at https://viz.stjude.cloud/community/cancer-survivorship-community-4/publications. Data relating to this work have been provided in a source data file. Source data are provided with this paper.

### Code availability

SAS codes, datasets, and a ReadMe file are available for download from https://zenodo.org/records/18396929.

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

## Acknowledgements

This work was supported by the National Cancer Institute (CA55727, G.T. Armstrong, Principal Investigator). Support to St. Jude Children's Research Hospital also provided by the Cancer Center Support (CORE) grant (CA21765, C. Roberts, Principal Investigator) and the American Lebanese-Syrian Associated Charities (ALSAC). AO was supported by grants from the Swedish Research Council (2022-00166 and 2024-02859), the Swedish Childhood Cancer Fund (PD2023-0003), and Gothenburg Medical Society (GLS-999210). The funders had no role in the planning, interpretation, or writing of the manuscript.

## Author contributions

We have read the Nature Portfolio journals´authorship policy. A.O., Q.L., S.I., S.B.D., C.S., P.C.N., Y.Y., G.T.A. & K.K.N. contributed significantly to the conception and the design of the study. A.O., Q.L., S.I., S.B.D., R.M.H., M.J.E., L.J., L.M.T., K.S., T.M.B., K.C.O., C.S., P.C.N., Y.Y., G.T.A. & K.K.N. contributed significantly to the acquisition, analysis, or interpretation of data. AO drafted the manuscript and all authors substantively revised it. All authors have approved the submitted version and have agreed both to be personally accountable for the author's own contributions and to ensure that questions related to the accuracy or integrity of any part of the work, even ones in which the author was not personally involved, are appropriately investigated, resolved, and the resolution documented in the literature.

## Funding

This work was supported by the National Cancer Institute (CA55727, G.T. Armstrong, Principal Investigator). Support to St. Jude Children's Research Hospital also provided by the Cancer Center Support (CORE) grant (CA21765, C. Roberts, Principal Investigator) and the American Lebanese-Syrian Associated Charities (ALSAC). AO was supported by grants from the Swedish Research Council (2022-00166 and 2024-02859), the Swedish Childhood Cancer Fund (PD2023-0003), Gothenburg Medical Society (GLS-999210), and Swedish governmental funding of clinical research (ALF). Open access funding provided by University of Gothenburg.

## Competing interests

All authors have completed the ICMJE conflicts of interest disclosure form. Q.L., L.J., L.M.T., D.S., T.M.B., K.C.O., P.C.N. and Y.Y., report no potential conflicts of interest. A.O. reports grants from the Swedish Research Council, the Swedish Childhood Cancer Fund, and Gothenburg Medical Society for the current study. A.O. reports travel grants from ASCO Conquer Cancer, AACR and the Swedish Childhood Cancer Fund for attending meetings. S.I. reports research funding from St. Jude Children's Research Hospital. S.B.D., G.T.A. and K.K.N. report funding from the National Institutes of Health and the National Cancer Institute to their institution. MJE reports research funding from Pfizer/Seagen, speaker honorarium from Amgen, and grant from COG outside the written work. RMH and CS report funding from the National Cancer Institute to their institutions through a subcontract from St. Jude Children´s Research Hospital for the current work. C.S. also reports research funding from Genentech and Pfizer, personal fees from Shionogi and Movember, and support from Shionogi and Executive Insight Healthcare Consultants for presentations at meetings outside the written work. The authors declare no other competing interests.
