## [Transparent Peer Review file · Nature Communications]

Potential for risk reduction of chronic health conditions through lifestyle in childhood cancer survivors

Corresponding Author: Dr Aron Onerup

Version 0:

Reviewer comments:

Reviewer #1

(Remarks to the Author)

Thank you for the opportunity to review the manuscript, "Potential for risk reduction of chronic health condition through lifestyle in childhood cancer survivors." This is an important, timely topic as pediatric cancer survivorship rates continue to grow and the need to identify modifiable long-term quality of life risk factors thereby increases. Overall, I appreciate the investigators' thoughtful approach to differentiating variance attributed to treatment factors (e.g., chemotherapy, radiotherapy) versus those attributable to non-treatment factors. Although I believe this paper stands to potentially substantively increment extant literature, some current concerns likely limit the manuscript's potential impact as currently written. Below, I have detailed my primary concerns:

1. This manuscript could likely be greatly clarified with overall re-organization. I encourage the authors to follow standard manuscript structure (abstract, introduction, methods section [which includes participants, measures, and procedures], data analytic strategy section, results, and discussion). As currently organized, particularly with the discussion coming before many methodological details are shared, it is challenging as reader to interpret the authors' findings in this important study.
2. I encourage the authors to provide greater detail about statistical methodology used in this study. For example, although the authors indicate that they are using 13-year longitudinal data from their participants, it is unclear how many time points they included during that time period, what psychometrically-sound assessment instruments were used, how data were treated (e.g., how specific indices were created), what variables were included in specific analyses, or what specific analyses were used for each hypothesis. Unfortunately, as written, this lack of clarity would not allow study replication and makes it challenging to fully appreciate or interpret findings. I encourage the authors to carefully revise the manuscript to ensure all required details are provided to allow study replication. As one step toward this goal, it would be helpful to enhance the data analytic strategy section with additional details about which analyses were selected (and how those analyses were completed, including which variables were included in each analysis) as well as justification for those specific analytic choices. Although the authors provide some details about the analytic strategy on page 12, the details provided do not provide clear rationale for statistical choices and are not sufficient to allow replication. Also particularly important in this revision will be defining how the investigators made decisions in their construction of the "healthy lifestyle" latent variable as well as how they defined and constructed the "mental QoL" variable given that anxiety and depression appear to have been included as separate constructs in at least some analyses (for example, see top of page 6 for inclusion of depression and anxiety outside of the larger mental QoL variable). Although some of this information is currently provided on pages 10-12, additional clarity would be needed to allow study replication. Of note, the authors indicate that previous studies have described some of the measures they used in this study; however, that statement alone does not clarify which specific measures were used in this study or how they were used. Lastly, in describing how and why the authors dichotomized or otherwise categorized self-reported behaviors, it is important to provide theory-driven and empirically-supported rationale for those specific categories and cut-offs (e.g., history of smoking at least 100 cigarettes as the threshold to determine whether someone engages in smoking behaviors). Where the authors choose to indicate that they based scoring on "international standards" (e.g., page 12), it is important to note to which standards they are referring.
3. Throughout the manuscript, I encourage the authors to ensure they are operationally defining all constructs that serve either as a study variable of interest or as rationale for the current study. For example, in the first paragraph of page one, the authors use the term "emotional health concerns" without defining whether they are referring to psychopathology, maladaptive coping behaviors, or other "emotional health" concerns/metrics. Similarly, throughout the manuscript, the authors seem to variably define "healthy lifestyle" such that the term sometimes includes healthy eating and physical activity and sometimes includes additional health behaviors (e.g., tobacco use). For all ambiguous constructs, I encourage the

authors to provide a citable, operational definition and ensure that any literature review about those constructs stay focused on studies that have operationalized the construct in the same way. Related to language specificity, I encourage the authors to carefully review the manuscript to ensure that all language is clear and direct, avoiding unnecessary ambiguity. For example, on the bottom of page two, rather than stating “components of the metabolic syndrome,” it would be helpful to list which (or at least some examples of) components. As another example, in the sentence, “The aims of this study among childhood cancer survivors were to...,” I encourage the authors to specify that the aims of this study were to evaluate these variable relations among adult survivors of pediatric cancer or to specify that they are interested in variable relations among individuals who completed cancer treatment at least XX years prior to study enrollment. Such language edits will help contextualize the study in a way that makes it easier for readers to interpret and situate among other literature.

4. The introduction, as currently written, implies that pediatric cancer survivors have elevated mental health risks; however, a robust literature has demonstrated that only a small subset of pediatric cancer survivors experience clinically elevated long-term mental health concerns. Although the subset who experience these concerns may experience greater symptom intensity than might otherwise be expected, the base rate for mental health disorders does not increase based on pediatric cancer history status.

5. The authors assert that previous evidence about links between healthy lifestyle behaviors and treatment-associated CHCs has been “low quality.” Why have those studies/findings been low quality, how are the authors defining low quality, and how will the current study rectify those limitations?

6. The authors note that their sample was demographically-similar across lifestyle groups. It would likely be helpful for a supplementary table to provide these demographics by lifestyle category, along with a definition of each lifestyle category and summary of statistical tests demonstrating that there were not significant differences between lifestyle groups across the demographic variables. Additionally, I would appreciate inclusion of cell size in each lifestyle/demographic group.

7. The authors also note that participants in the unhealthy lifestyle group were less likely to have a college education, high income, or health insurance and more likely to be married than those with a healthy lifestyle. These are major confounds in this study and should be both statistically controlled and more fully discussed in the discussion. For example, lack of health insurance and limited financial resources, combined with increased stress related to low socioeconomic status, could have a sizeable impact on CHC development. It would therefore be problematic to infer that healthy lifestyle behaviors, specifically, explained differences in CHC development when this much broader set of risk factors also systematically differed by lifestyle classification.

8. The use of siblings as a control group could be problematic in a study like this because of likely overlap between patient and sibling lifestyle history and genetic predispositions but also because of siblings’ intentional sibling differentiation that often occurs during one child’s pediatric cancer treatment. There are many reasons siblings could have exaggerated similarity to patients (e.g., biological and historical environmental factors) as well as reasons that siblings could intentionally choose specific lifestyle behaviors based on their experiences with their brother’s or sister’s cancer treatment. Although I recognize that siblings have long been used as a convenient healthy control group in pediatric cancer long-term follow-up studies, it is important to fully consider ways in which siblings differ from true healthy population controls when investigators are interpreting their findings.

9. Throughout the discussion, especially in light of study limitations, I encourage the authors to carefully review their text to ensure findings are not over-stated. This is particularly important for any statements that imply causality in this non-experimental study.

10. Please ensure appropriate citations are provided for all factual statements throughout the manuscript, especially throughout the discussion. For example, on page 7, the text indicates that healthy lifestyle interventions are often resource intensive and may not be effective for pediatric cancer survivors, but no citations are provided for these factual statements.

11. I recommend the authors clarify how the following statement applies to the current study: “This enabled longitudinal follow-up, removing the risk of reverse causality seen in cross-sectional studies of lifestyle and CHCs.” Cancer, itself, is a chronic health condition and this study did not use an experimental design, so it is unclear how causality itself could be inferred from these findings. This is especially true given the important confounds outlined in the paper and in this review. In addition to the concerns above, I also recommend the authors consider the following minor considerations:

- As a more minor point, I recommend the authors carefully review the paper for language efficiency, so as to remove any content that is not necessary to justify, explain, or interpret for the study. For example, on page one, the first sentence of the second paragraph does not appear to add substantive context for the current study, does not provide a thesis for the paragraph, and could likely be removed without detriment to the overall paper. As another example, the authors present both European and U.S. literature in the introduction despite that the study appears to only use U.S. participants. Lifestyle behaviors are largely rooted in culture, so findings outside the U.S. are likely not pertinent in setting up justification for this study. Further, related to language specificity, it will be important to carefully extract language that suggests causality in this non-experimental study (e.g., page 5, first sentence of last paragraph).

- On page 4, the authors use the phrase “borderline statistically significant” in their presentation of findings. First, it will be important for the authors to clearly define their criteria for statistical significance, including whether they employed any correction for the number of analyses. Once the threshold for statistical significance is clarified, it is important to frame the findings as either statistically significant or not. Even more importantly than presenting statistical significance, it will be important for the authors to present effect sizes and to describe clinical significance of the magnitude of any findings.

- Where language in the manuscript asserts that multiple studies have documented a phenomenon, it would be helpful for the authors to either cite multiple studies or a meta-analysis/systematic review that demonstrates the finding has been documented more than once. For example, on page 8, the authors say that “previous reports” demonstrate that healthy lifestyle interventions are not always effective for pediatric cancer survivors; however, they provide only one study-specific citation to support this assertion. Additionally, the study they cite in this specific example was an intervention only for tobacco cessation and did not encompass other healthy lifestyle behaviors that are included in this study.

- I recommend that the authors review the text to ensure all sentence fragments are corrected (e.g., page 8, “Since physical activity and diet are more effective at preventing than treating obesity in the general population.”).

Reviewer #2

(Remarks to the Author)

This manuscript addresses an important public health topic, the attributable risk of lifestyle to chronic health conditions among adult childhood cancer survivors (CCS). Although a robust body of literature on lifestyle and health outcomes exists for this population, the results provided in this study go beyond the current evidence. The authors provide a sufficient review of the literature, highlighting the critical gaps addressed by the current study.

The study relies on the Childhood Cancer Survivor Study, a large longitudinal cohort study that has a well-characterized population and contains a robust dataset including information on diagnostic, treatment, lifestyle behavior, psychosocial factors, health history and other variables. Overall, the approach to the study is well documented and reflects rigor. Data tables and figures are well-labeled, and visualization makes interpretation easy to follow. Use of time-varying lifestyle exposures updated longitudinally, with lagging to reduce reverse causality is a strength. As is the comparison to sibling controls. Interpretation of the results is sound.

Study findings represent an important contribution to the literature, showing that lifestyle behaviors, compared with chemo and radiotherapy, are significantly associated with physical and emotional health. These results challenge the assumption that treatments are the primary driver of adverse physical and mental health outcomes among adult CCS. Comparisons to sibling controls provide additional context by showing similarities and differences in risks between these two populations. Identifying the primary behavioral contributors (obesity, physical activity) will inform targets for future interventions. In sum, the study provides robust evidence to further reinforce the message that lifestyle matters. The discussion is well-considered, acknowledges the uniqueness of this study and the data set, as well as limitations and provides valuable insight into potential next steps.

Areas needing attention include:

While it is appreciated that the CCSS methods are published elsewhere, further details on the CCSS questionnaires are needed to fully understand the results and any potential limitations for this study. The link to the St. Jude website is not helpful. Instead, it is suggested that information on how often questionnaires were completed (every 5 years?) and how many questionnaires participants completed on average.

It is not clear in the methods section if CHCs were self-reported?

Details on what is meant by time-dependent SES was calculated – was this updated at each questionnaire?

Study limitations. The discussion section provides a good review of study limitations, although further attention to the lack of dietary data is warranted. Additionally, there should be some acknowledgement of the higher potential for Type 1 error, given the decision to not correct for multiple comparisons.

Additional (minor editorial comments)

Lines 93-96: awkward phrasing that makes the sentence hard to understand – suggest – “Those with a higher prevalence of unhealthy lifestyle factors were less likely to have a (the a is missing) college education.....”

Line 97: It is not clear what is meant by events and each outcome. Using the name of the table would clarify this confusion, “Number of events by lifestyle score category.”

Reviewer #3

(Remarks to the Author)

The submitted manuscript presents novel and compelling findings, making a valuable contribution to the field of cancer epidemiology. The analysis is grounded in a robust national dataset, and the work is well-conceived and thoughtfully executed.

The absence of a power calculation is a notable omission. Including such an analysis would strengthen the manuscript by clarifying the anticipated statistical power and reinforcing the validity of the findings.

The primary exposure of interest, ‘lifestyle’, is both relevant and impactful. The dataset offers rich information for capturing time-varying lifestyle behaviours. While the authors have understandably opted for a simplified and heavily categorized measure to facilitate reporting and interpretation, it remains the central exposure. It would be beneficial to include a more granular definition or an alternative operationalization of lifestyle as part of a sensitivity analysis.

Table 1 presents a cross-tabulation by lifestyle categories. The authors should clarify how these categories were assigned, particularly in cases where lifestyle varies over time. It is important to specify the time point or period used to define lifestyle in this context.

The authors have appropriately adjusted for age using a cubic spline and included an interaction with lifestyle, which reflects good epidemiological practice given the age-related nature of chronic health conditions. Providing additional detail on these

modeling choices would enhance transparency. While further interactions, such as age-sex-lifestyle, could offer deeper insights, it is reasonable that the scope of the analysis was limited.

Household income is presented in categorized form within the tables. It would be helpful to confirm whether this categorization was also applied during model adjustment.

The use of Generalized Estimating Equations (GEE) is methodologically sound. Nonetheless, the limitations of GEE, particularly its assumption of data being missing completely at random (MCAR) and the challenges it poses for model comparison, should be acknowledged and discussed.

The specification of the time-varying lifestyle score is clearly articulated. Was any sensitivity analysis conducted to assess the robustness of this measure?

Finally, since participants could respond on multiple occasions, it would be informative to include the total number of responses per participant in Table 1. This could help assess potential concerns related to response patterns and the validity of the MCAR assumption.

Version 1:

Reviewer comments:

Reviewer #1

(Remarks to the Author)

Thank you for the opportunity to re-review this manuscript. As I previously stated, this manuscript reflects a topic of critical importance to the field and could serve as a springboard to support future efforts to improve childhood cancer survivors' healthy lifestyle behaviors. I appreciate the authors' keen attention to the previous reviews and believe the manuscript has been significantly enhanced with the authors' revisions. Please see below for my remaining recommendations to continue enhancing the strength of this paper:

- It would be helpful for the limitation section to further describe the significant limitation of using siblings as a control group for adults who are pediatric cancer survivors. Although the CCSS is a valuable resource and, arguably, one of the most robust, impressive longitudinal datasets available to pediatric cancer survivorship researchers, there remain significant confounds of using siblings as a control group. In some ways, siblings are too much alike to provide a population-esque control—that is, siblings often share many of the same social, biological, and environmental exposures irrespective of one child's chronic illness. On the other hand, sibling dyads in which one child has been treated for cancer also have many more distinctions than merely one child's cancer treatment. For example, parents, peers, and other family members often treat siblings differently during and following one child's cancer treatment. Pertaining to healthy lifestyles, pediatric cancer survivors may also have real or perceived differences in behavioral expectations than their healthy siblings. It would be helpful for the authors to consider the potential role of the sibling control group in a more nuanced way, particularly how the use of a sibling control group could have influenced this study's findings, specifically.
- The first half of the discussion, as currently written, reads a bit like a restatement of the results. Rather than merely repeatedly stating that this study is the first of its kind, it could be helpful for the authors to more clearly describe how this study's findings fit within existing theoretical frameworks to understand childhood cancer survivors' long-term lifestyle-related risks.
- On the top of page 8, the authors state, "...may be more cost-effective in childhood survivors than in the general population if we assume a similar effectiveness for behavior change..." However, it is likely not appropriate to assume that healthy lifestyle interventions designed for healthy populations would have the same effectiveness for pediatric cancer survivors. I encourage the authors to consider rewording.
- As written, the discussion seems to imply that researchers have yet to develop healthy lifestyle interventions that specifically target adolescents/young adults or young adults who are pediatric cancer survivors. However, several such interventions exist and continue to be tested in current literature. It could be helpful for the authors to specifically (albeit briefly) either provide some examples of such interventions and/or describe what the limitations of such interventions are (and how the current study's findings suggest the existing interventions should be strengthened). For example, although some recent healthy lifestyle interventions for AYA/YA pediatric cancer survivors have begun focusing on limiting attrition and improving participant engagement, many previous interventions demonstrated relatively high attrition.
- On page 8, the authors suggest that many previous healthy lifestyle interventions have not achieved "fully healthy lifestyles." What does this phrase mean, operationally?
- On page 7, lines 175-176, it is unclear what the authors mean when they state, "Our results now substantiate previous reports that have suggested associations between a healthy lifestyle and reduced risk for late mortality." Wouldn't late mortality (as opposed to death at a young age) be a good thing? Hasn't most extant literature shown that healthy lifestyles do, in fact, tend to delay mortality into later adulthood?
- In the limitations section, the authors rightfully report that BMI is not an appropriate proxy for healthy eating behaviors because BMI is impacted by physical activity, biological diathesis, and eating behaviors. However, BMI could also be impacted by cancer treatment-related late-effects, like metabolic changes. Because of the nature of the study sample, I encourage the authors to reflect the potential impact of cancer treatment (or even tumor location in the case of some brain tumors) on BMI.
- On page 8, I recommend breaking the text into multiple, shorter paragraphs to reduce cognitive load for the reader. For example, the authors might consider beginning new paragraphs at "However, BMI reflects both diet..." and at "Both the exposures and outcomes in our study were self-reported..."

- The authors state that causality can be implied by their study's longitudinal design; however, because causality can only be inferred from experimental design, it is overstated to say that even a large-sample, longitudinal study can imply causality.

(Remarks on code availability)

No comments

Reviewer #2

(Remarks to the Author)

The revised manuscript is very responsive to earlier critiques and will make a significant contribution to the literature on childhood cancer survivorship.

(Remarks on code availability)

Reviewer #3

(Remarks to the Author)

I am satisfied by the responses given by the authors and the amendments to their manuscript.

(Remarks on code availability)

Reviewer expertise:

- Reviewer 1: lifestyle and outcomes in childhood cancer survivors, psychological effects
- Reviewer 2: lifestyle and outcomes in childhood cancer survivors, epidemiology, cancer prevention, cancer/ health disparities
- Reviewer 3: biostatistics, health research

REVIEWER COMMENTS

Reviewer #1 (Remarks to the Author):

Thank you for the opportunity to review the manuscript, “Potential for risk reduction of chronic health condition through lifestyle in childhood cancer survivors.” This is an important, timely topic as pediatric cancer survivorship rates continue to grow and the need to identify modifiable long-term quality of life risk factors thereby increases. Overall, I appreciate the investigators’ thoughtful approach to differentiating variance attributed to treatment factors (e.g., chemotherapy, radiotherapy) versus those attributable to non-treatment factors. Although I believe this paper stands to potentially substantively increment extant literature, some current concerns likely limit the manuscript’s potential impact as currently written. Below, I have detailed my primary concerns:

Response: *We appreciate the overall assessment of this manuscript.*

1. This manuscript could likely be greatly clarified with overall re-organization. I encourage the authors to follow standard manuscript structure (abstract, introduction, methods section [which includes participants, measures, and procedures], data analytic strategy section, results, and discussion). As currently organized, particularly with the discussion coming before many methodological details are shared, it is challenging as reader to interpret the authors’ findings in this important study.

Response: *We understand the wish for standard manuscript structure. However, the manuscript was formatted according to the formatting requirements from Nature Communications (<https://www.nature.com/ncomms/submit/article>), with a format of a brief abstract and methods coming after the discussion.*

2. I encourage the authors to provide greater detail about statistical methodology used in this study. For example, although the authors indicate that they are using 13-year longitudinal data from their participants, it is unclear how many time points they included during that time period, what psychometrically-sound assessment instruments were used, how data were treated (e.g., how specific indices were created), what variables were included in specific analyses, or what specific analyses were used for each hypothesis. Unfortunately, as written, this lack of clarity would not allow study replication and makes it challenging to fully appreciate or interpret findings. I encourage the authors to carefully revise the manuscript to ensure all required details are provided to allow study replication. As one step toward this goal, it would be helpful to enhance the data analytic strategy section with additional details about which analyses were selected (and how those analyses were completed, including

which variables were included in each analysis) as well as justification for those specific analytic choices. Although the authors provide some details about the analytic strategy on page 12, the details provided do not provide clear rationale for statistical choices and are not sufficient to allow replication. Also particularly important in this revision will be defining how the investigators made decisions in their construction of the “healthy lifestyle” latent variable as well as how they defined and constructed the “mental QoL” variable given that anxiety and depression appear to have been included as separate constructs in at least some analyses (for example, see top of page 6 for inclusion of depression and anxiety outside of the larger mental QoL variable). Although some of this information is currently provided on pages 10-12, additional clarity would be needed to allow study replication. Of note, the authors indicate that previous studies have described some of the measures they used in this study; however, that statement alone does not clarify which specific measures were used in this study or how they were used. Lastly, in describing how and why the authors dichotomized or otherwise categorized self-reported behaviors, it is important to provide theory-driven and empirically-supported rationale for those specific categories and cut-offs (e.g., history of smoking at least 100 cigarettes as the threshold to determine whether someone engages in smoking behaviors). Where the authors choose to indicate that they based scoring on “international standards” (e.g., page 12), it is important to note to which standards they are referring.

Response: *We have now revised the methods section to include all requested information. For the number of assessments per participants, this has now been included in table 1, as also requested by reviewers 2 and 3. As shown in table 1, the mean number of assessments was 4 with a range of 2-7. For the categories of the exposures used in the analyses, these are described under “exposures”. We have now added references to where these were created. For the outcome measures, these are described under “outcomes”. For the psychometric measures, these were BSI-18 and SF-36 and categorization was done as described under “outcomes”. For the lifestyle score, this composite score was created in a previous publication from the CCSS (Dixon et al., Lancet, 2023). However, in our study overweight resulted in a 0.5 point addition to the lifestyle score to reflect the dose-response association between body weight and CHC. This is now clarified in the methods section. The mental QoL variable is the same as the mental component summary scale of the SF-36 and this has now been clarified in the methods.*

3. Throughout the manuscript, I encourage the authors to ensure they are operationally defining all constructs that serve either as a study variable of interest or as rationale for the current study. For example, in the first paragraph of page one, the authors use the term “emotional health concerns” without defining whether they are referring to psychopathology, maladaptive coping behaviors, or other “emotional health” concerns/metrics. Similarly, throughout the manuscript, the authors seem to variably define “healthy lifestyle” such that the term sometimes includes healthy eating and physical activity and sometimes includes additional health behaviors (e.g., tobacco use). For all ambiguous constructs, I encourage the authors to provide a citable, operational definition and ensure that any literature review about those constructs stay focused on studies that have operationalized the construct in the same way. Related to language specificity, I encourage the authors to carefully review the manuscript to ensure that all language is clear and direct, avoiding unnecessary ambiguity. For example, on the bottom of page two, rather than stating “components of the metabolic syndrome,” it would be helpful to list which (or at least some examples of) components. As

another example, in the sentence, “The aims of this study among childhood cancer survivors were to...,” I encourage the authors to specify that the aims of this study were to evaluate these variable relations among adult survivors of pediatric cancer or to specify that they are interested in variable relations among individuals who completed cancer treatment at least XX years prior to study enrollment. Such language edits will help contextualize the study in a way that makes it easier for readers to interpret and situate among other literature.

Response: *We have now changed the wording in the first section relating to emotional health to stress that this relates to symptoms. We have also added references for both emotional health symptoms and QoL to this statement.*

For this manuscript, we defined lifestyle as including the absence of smoking and heavy/risky alcohol consumption, maintaining a healthy weight, and engaging in physical activity, as defined in the first section of the introduction. This is also what we included in our lifestyle score. As several previous studies have focused on individual of these lifestyle factors, some references in our literature review will provide information for some but not all of these lifestyle factors. As our study is an original study and not a systematic review, space limitations prevent us from systematically presenting the previously published results for each lifestyle factor with each CHC. However, there is a coming systematic review from the International Guidelines Harmonization Group (IGHG) for this specific purpose, which is soon to be published. Therefore, our literature review presents the previous results for any or all of these factors with the outcomes used in our study. We have now gone through the text and clarified according to the suggestions in relevant places.

4. The introduction, as currently written, implies that pediatric cancer survivors have elevated mental health risks; however, a robust literature has demonstrated that only a small subset of pediatric cancer survivors experience clinically elevated long-term mental health concerns. Although the subset who experience these concerns may experience greater symptom intensity than might otherwise be expected, the base rate for mental health disorders does not increase based on pediatric cancer history status.

Response: *We have now changed the wording in the introduction to clarify that we are referring to emotional health symptoms of anxiety and depression and provided references for both these and QoL. However, the focus of this study is not the comparison with the general population but with the causes for experiencing the outcomes during adult survivorship. Interestingly, our study shows that while the risk of developing each outcome is higher in specific subsets of the overall childhood cancer survivorship population, the AER for developing anxiety, depression, or impaired QoL is significantly higher in the overall survivor population than in sibling controls.*

5. The authors assert that previous evidence about links between healthy lifestyle behaviors and treatment-associated CHCs has been “low quality.” Why have those studies/findings been low quality, how are the authors defining low quality, and how will the current study rectify those limitations?

Response: *While this may sound like a judgment from our side, we were referring to the quality assessment performed according to the established GRADE criteria in the systematic review that we referenced.*

6. The authors note that their sample was demographically-similar across lifestyle groups. It would likely be helpful for a supplementary table to provide these demographics by lifestyle category, along with a definition of each lifestyle category and summary of statistical tests demonstrating that there were not significant differences between lifestyle groups across the demographic variables. Additionally, I would appreciate inclusion of cell size in each lifestyle/demographic group.

Response: *Table 1 provides the demographics by lifestyle category. We have now clarified it by providing the coding of the lifestyle score. However, differences in baseline characteristics were not tested, as Table 1 is for descriptive purposes only. That is, provision of Table 1's data are for understanding the study population, and it is not intended for statistical inference or hypothesis testing. Note that, in studies with large sample sizes such as the CCSS, even trivial differences in baseline characteristics can yield statistical significance that lacks clinical relevance. Therefore, in large cohorts, p values in Table 1 are often impacted by the large sample size, and their inclusion may mislead readers into overinterpreting minor differences as meaningful. Importantly, while the distributions of sex, age, and race/ethnicity were comparable across groups, these were nonetheless included as covariates in the models as these had been identified as possible confounders in the DAG.*

7. The authors also note that participants in the unhealthy lifestyle group were less likely to have a college education, high income, or health insurance and more likely to be married than those with a healthy lifestyle. These are major confounds in this study and should be both statistically controlled and more fully discussed in the discussion. For example, lack of health insurance and limited financial resources, combined with increased stress related to low socioeconomic status, could have a sizeable impact on CHC development. It would therefore be problematic to infer that healthy lifestyle behaviors, specifically, explained differences in CHC development when this much broader set of risk factors also systematically differed by lifestyle classification.

Response: *We agree. As this was expected, these variables were included as covariates in the model, as described in the statistics section and in the figure captions for figures 1 and 4:*

"All analyses were adjusted for potential confounders, selected based on a hypothesized directed acyclic graph: attained age as cubic splines, sex, race/ethnicity, age at cancer diagnosis, time-dependent socioeconomic status (SES) including education, marriage status, household income, and insurance, and treatment exposures."

8. The use of siblings as a control group could be problematic in a study like this because of likely overlap between patient and sibling lifestyle history and genetic predispositions but also because of siblings' intentional sibling differentiation that often occurs during one child's pediatric cancer treatment. There are many reasons siblings could have exaggerated similarity to patients (e.g., biological and historical environmental factors) as well as reasons that siblings could intentionally choose specific lifestyle behaviors based on their experiences with their brother's or sister's cancer treatment. Although I recognize that siblings have long been used as a convenient healthy control group in pediatric cancer long-term follow-up studies, it is important to fully consider ways in which siblings differ from true healthy population controls when investigators are interpreting their findings.

Response: *We agree that the lived experiences of siblings of childhood cancer survivors may differentiate them from the general population. This may be the case especially for emotional distress. However, in our study we use a non-survivor control group, siblings, only for the analyses of the AER for having an unfavorable lifestyle. While the distribution of the lifestyle factors and potentially for some of the outcomes may differ between siblings and the general population, the individual absolute excess risk associated with having less than optimal lifestyle should not differ. Second, if the AER in siblings is more similar to the AER in survivors than in the general population, that would mean that the differences between survivors and siblings seen in our study underestimate the actual difference between survivors and the general population. Third, having shared genetic and lifestyle history would mean that the difference observed between the two groups is explained mainly by modification of treatment exposure. We have added the following sentence to the limitations section in the discussion:*

“The use of siblings as control group for the comparison of AER is a limitation as these may not represent the general population in some respects. However, the absolute excess risks should not be affected by this.”

9. Throughout the discussion, especially in light of study limitations, I encourage the authors to carefully review their text to ensure findings are not over-stated. This is particularly important for any statements that imply causality in this non-experimental study.

Response: *We agree with this need and have gone through the discussion to remove any phrasing that could suggest causality.*

10. Please ensure appropriate citations are provided for all factual statements throughout the manuscript, especially throughout the discussion. For example, on page 7, the text indicates that healthy lifestyle interventions are often resource intensive and may not be effective for pediatric cancer survivors, but no citations are provided for these factual statements.

Response: *We have gone through the manuscript to ensure references for all factual statements.*

11. I recommend the authors clarify how the following statement applies to the current study: “This enabled longitudinal follow-up, removing the risk of reverse causality seen in cross-sectional studies of lifestyle and CHCs.” Cancer, itself, is a chronic health condition and this study did not use an experimental design, so it is unclear how causality itself could be inferred from these findings. This is especially true given the important confounds outlined in the paper and in this review.

Response: *The emphasis in the cited sentence should be on **reverse**. In a cross-sectional study, the associations observed in our study could very well have represented reverse pathways – i.e., that some CHCs lead to worse lifestyle behaviors. For some associations, that could represent a majority of associations observed, not least for associations between emotional distress and some lifestyle behaviors. Therefore, while our observational study cannot claim causality, the temporality of the associations is known, strengthening the hypothesis that the associations observed might reflect causal relations.*

In addition to the concerns above, I also recommend the authors consider the following minor considerations:

- As a more minor point, I recommend the authors carefully review the paper for language efficiency, so as to remove any content that is not necessary to justify, explain, or interpret for the study. For example, on page one, the first sentence of the second paragraph does not appear to add substantive context for the current study, does not provide a thesis for the paragraph, and could likely be removed without detriment to the overall paper. As another example, the authors present both European and U.S. literature in the introduction despite that the study appears to only use U.S. participants. Lifestyle behaviors are largely rooted in culture, so findings outside the U.S. are likely not pertinent in setting up justification for this study. Further, related to language specificity, it will be important to carefully extract language that suggests causality in this non-experimental study (e.g., page 5, first sentence of last paragraph).

Response: *We have revised the language of the manuscript to remove any unnecessary sentences and remove any language suggesting causality.*

- On page 4, the authors use the phrase “borderline statistically significant” in their presentation of findings. First, it will be important for the authors to clearly define their criteria for statistical significance, including whether they employed any correction for the number of analyses. Once the threshold for statistical significance is clarified, it is important to frame the findings as either statistically significant or not. Even more importantly than presenting statistical significance, it will be important for the authors to present effect sizes and to describe clinical significance of the magnitude of any findings.

Response: *We thank the reviewer for pointing out this and we have now removed it. We have also clarified the significance level used (5%) in the statistics section.*

- Where language in the manuscript asserts that multiple studies have documented a phenomenon, it would be helpful for the authors to either cite multiple studies or a meta-analysis/systematic review that demonstrates the finding has been documented more than once. For example, on page 8, the authors say that “previous reports” demonstrate that healthy lifestyle interventions are not always effective for pediatric cancer survivors; however, they provide only one study-specific citation to support this assertion. Additionally, the study they cite in this specific example was an intervention only for tobacco cessation and did not encompass other healthy lifestyle behaviors that are included in this study.

Response: *We agree. We have now gone through the manuscript to align the language with the underlying evidence. For the specific sentence, it now reads: “Unfortunately, validated lifestyle interventions may not be effective in childhood cancer survivors despite being effective in the general population.”*

- I recommend that the authors review the text to ensure all sentence fragments are corrected (e.g., page 8, “Since physical activity and diet are more effective at preventing than treating obesity in the general population.”).

Response: *We have performed an extensive language check.*

Reviewer #2 (Remarks to the Author):

This manuscript addresses an important public health topic, the attributable risk of lifestyle to chronic health conditions among adult childhood cancer survivors (CCS). Although a robust body of literature on lifestyle and health outcomes exists for this population, the results provided in this study go beyond the current evidence. The authors provide a sufficient review of the literature, highlighting the critical gaps addressed by the current study.

The study relies on the Childhood Cancer Survivor Study, a large longitudinal cohort study that has a well-characterized population and contains a robust dataset including information on diagnostic, treatment, lifestyle behavior, psychosocial factors, health history and other variables. Overall, the approach to the study is well documented and reflects rigor. Data tables and figures are well-labeled, and visualization makes interpretation easy to follow. Use of time-varying lifestyle exposures updated longitudinally, with lagging to reduce reverse causality is a strength. As is the comparison to sibling controls. Interpretation of the results is sound.

Study findings represent an important contribution to the literature, showing that lifestyle behaviors, compared with chemo and radiotherapy, are significantly associated with physical and emotional health. These results challenge the assumption that treatments are the primary driver of adverse physical and mental health outcomes among adult CCS. Comparisons to sibling controls provide additional context by showing similarities and differences in risks between these two populations. Identifying the primary behavioral contributors (obesity, physical activity) will inform targets for future interventions. In sum, the study provides robust evidence to further reinforce the message that lifestyle matters. The discussion is well-considered, acknowledges the uniqueness of this study and the data set, as well as limitations and provides valuable insight into potential next steps.

Response: *We would like to thank the reviewer for taking your time to read the manuscript and for the overall assessment of our study.*

Areas needing attention include:

While it is appreciated that the CCSS methods are published elsewhere, further details on the CCSS questionnaires are needed to fully understand the results and any potential limitations for this study. The link to the St. Jude website is not helpful. Instead, it is suggested that information on how often questionnaires were completed (every 5 years?) and how many questionnaires participants completed on average.

Response: *We agree, and this was also suggested by reviewers 1 and 3. The information on number of questionnaires per participant is now available in table 1. The mean number of questionnaires per participant was 4, with a range of 2-7.*

It is not clear in the methods section if CHCs were self-reported?

Response: *The CHCs were self-reported. To clarify this, we modified the first sentence describing the outcomes: “The time of first occurrence of each CHC over time was assessed by a series of questions in every study questionnaire...”*

Details on what is meant by time-dependent SES was calculated – was this updated at each questionnaire?

Response: *Yes, SES was updated during follow-up. To clarify this, we have included the term time-dependent in the sentence describing how the lifestyle variables were updated during follow-up to show that this applies also to SES where it is described as time-dependent.*

Study limitations. The discussion section provides a good review of study limitations, although further attention to the lack of dietary data is warranted. Additionally, there should be some acknowledgement of the higher potential for Type 1 error, given the decision to not correct for multiple comparisons.

Response: *We have added the two following sentences to the discussion:*

“However, the lack of dietary data means that our study provides no information on associations between specific types of diet and CHCs.”

“As we considered all CHCs to represent individual hypotheses, we did not adjust for multiple comparisons. This may increase risk for Type 1 error, and should be taken into account when interpreting our results.”

Additional (minor editorial comments)

Lines 93-96: awkward phrasing that makes the sentence hard to understand – suggest – “Those with a higher prevalence of unhealthy lifestyle factors were less likely to have a (the a is missing) college education.....”

Response: *Thank you, we now changed to:*

“Participants with an unhealthy lifestyle score were less likely to have college education, a high income, and health insurance, and more likely to be married than those with a healthy lifestyle.”

Line 97: It is not clear what is meant by events and each outcome. Using the name of the table would clarify this confusion, “Number of events by lifestyle score category.”

Response: *We changed according to your suggestion.*

Reviewer #3 (Remarks to the Author):

The submitted manuscript presents novel and compelling findings, making a valuable contribution to the field of cancer epidemiology. The analysis is grounded in a robust national dataset, and the work is well-conceived and thoughtfully executed.

Response: *We would like to thank the reviewer for taking your time to critically read and suggest improvements to our manuscript.*

The absence of a power calculation is a notable omission. Including such an analysis would strengthen the manuscript by clarifying the anticipated statistical power and reinforcing the validity of the findings.

Response: *While having a power calculation might be a strength, it has been argued with emphasis that low power should not refrain from performing observational analyses in existing databases (Hernan, J Clin Epidemiol, 2022). The case for planning intervention studies is of course something completely different, where it is unethical to have participants undergo study-specific interventions if the study is not large enough to detect the effect. In addition to this, the large number of primary outcomes would lead to a need for several sample size estimations. Therefore, we decided a priori to refrain from performing any power calculation. We have clarified the statement on this in the statistics section:*

“No formal sample size/power calculation was performed as this was an analysis of an existing database and there were several primary outcomes.”

The primary exposure of interest, ‘lifestyle’, is both relevant and impactful. The dataset offers rich information for capturing time-varying lifestyle behaviours. While the authors have understandably opted for a simplified and heavily categorized measure to facilitate reporting and interpretation, it remains the central exposure. It would be beneficial to include a more granular definition or an alternative operationalization of lifestyle as part of a sensitivity analysis.

Response: *We agree that the primary exposure used is crude and that a more granular definition might help interpret the results. However, we consider our analyses of the underlying individual lifestyle variables to provide this granularity and in our opinion, this provides more useful information than a more granular composite exposure. Combined, the two analyses can provide information on the combined potential of lifestyle and on the specific lifestyle behaviors of interest for preventing each CHC.*

Table 1 presents a cross-tabulation by lifestyle categories. The authors should clarify how these categories were assigned, particularly in cases where lifestyle varies over time. It is important to specify the time point or period used to define lifestyle in this context.

Response: *Thank you for pointing out this need for clarification. We have now added “at baseline” to the heading. And in the table footnote, we include the definition of the lifestyle score as described in the methods.*

The authors have appropriately adjusted for age using a cubic spline and included an interaction with lifestyle, which reflects good epidemiological practice given the age-

related nature of chronic health conditions. Providing additional detail on these modeling choices would enhance transparency. While further interactions, such as age-sex-lifestyle, could offer deeper insights, it is reasonable that the scope of the analysis was limited.

Response: *We agree that further clarification of our modeling choices can improve transparency. In the revised manuscript, we have expanded the Methods section to specify that attained age was modeled using restricted cubic splines with five knots placed at the 5th, 25th, 50th, 75th, and 95th percentiles of the age distribution. This flexible specification allowed for non-linear associations between age and each outcome. Interaction terms between the lifestyle score and attained age were included to assess whether the association between lifestyle and chronic health conditions varied across age. The interaction term was tested using likelihood ratio tests, and statistically significant interactions were reported in Supplementary Table 2. No higher-order interactions (e.g., age–sex–lifestyle) were included, as they were beyond the prespecified analytic scope. The following sentences were added to the methods:*

“Attained age was modeled using restricted cubic splines with five knots placed at the 5th, 25th, 50th, 75th, and 95th percentiles of the age distribution to allow for non-linear associations with each outcome. To evaluate whether associations between lifestyle and outcomes differed across age, interaction terms between attained age and the lifestyle score were assessed using likelihood ratio tests.”

Household income is presented in categorized form within the tables. It would be helpful to confirm whether this categorization was also applied during model adjustment.

Response: *We have now clarified the categorization of SES in the statistics section: “Socioeconomic status was categorized as in Table 1.”*

The use of Generalized Estimating Equations (GEE) is methodologically sound. Nonetheless, the limitations of GEE, particularly its assumption of data being missing completely at random (MCAR) and the challenges it poses for model comparison, should be acknowledged and discussed.

Response: *We agree that Generalized Estimating Equations (GEE) have certain limitations, particularly regarding assumptions about missing data and challenges in comparing model fit. We have therefore added a statement in the Discussion acknowledging that our GEE analyses assume data to be missing completely at random (MCAR) and that model comparison is less straightforward than in likelihood-based approaches. We added the following to the strengths and limitations section of the discussion:*

“In the analyses of emotional distress and quality of life, we used generalized estimating equations (GEE) to account for within-individual correlation across repeated measurements. While GEE provides robust population-averaged estimates, it assumes that data are missing completely at random and does not provide a likelihood function, limiting formal model comparison.”

The specification of the time-varying lifestyle score is clearly articulated. Was any sensitivity analysis conducted to assess the robustness of this measure?

Response: We thank the reviewer for this insightful comment. No separate sensitivity analyses were conducted for the time-varying lifestyle score. However, robustness was addressed through several design choices consistent with the original CCSS mortality analysis by Dixon et al. (Lancet, 2023). In that study, and in our analysis, lifestyle was modelled as a time-varying exposure updated at each questionnaire, with changes implemented at midpoints between assessments. Missing values were handled by last observation carried forward or mean substitution when at least two lifestyle components were available, and all exposures were lagged five years before the outcome to minimize reverse causality. These procedures were prespecified to enhance robustness of the time-varying measure and were therefore considered sufficient in place of a separate sensitivity analysis.

Finally, since participants could respond on multiple occasions, it would be informative to include the total number of responses per participant in Table 1. This could help assess potential concerns related to response patterns and the validity of the MCAR assumption.

Response: We agree and have now added this information to Table 1. The mean number of responses per participant was 4 with a range of 2-7.

Reviewer 1

Thank you for the opportunity to re-review this manuscript. As I previously stated, this manuscript reflects a topic of critical importance to the field and could serve as a springboard to support future efforts to improve childhood cancer survivors' healthy lifestyle behaviors. I appreciate the authors' keen attention to the previous reviews and believe the manuscript has been significantly enhanced with the authors' revisions. Please see below for my remaining recommendations to continue enhancing the strength of this paper:

1. It would be helpful for the limitation section to further describe the significant limitation of using siblings as a control group for adults who are pediatric cancer survivors. Although the CCSS is a valuable resource and, arguably, one of the most robust, impressive longitudinal datasets available to pediatric cancer survivorship researchers, there remain significant confounds of using siblings as a control group. In some ways, siblings are too much alike to provide a population-esque control—that is, siblings often share many of the same social, biological, and environmental exposures irrespective of one child's chronic illness. On the other hand, sibling dyads in which one child has been treated for cancer also have many more distinctions than merely one child's cancer treatment. For example, parents, peers, and other family members often treat siblings differently during and following one child's cancer treatment. Pertaining to healthy lifestyles, pediatric cancer survivors may also have real or perceived differences in behavioral expectations than their healthy siblings. It would be helpful for the authors to consider the potential role of the sibling control group in a more nuanced way, particularly how the use of a sibling control group could have influenced this study's findings, specifically.

Response: We agree that sibling comparators may differ from the general population. Depending on the purpose of the study, this may be both a strength or a limitation, as recently discussed in an article in JAMA (10.1001/jama.2025.22234). Following the reviewer's suggestion and clarification, we have modified the section on sibling comparators in the limitations section:

“The use of siblings as a reference group for the comparison of AER is a limitation. Siblings may not represent the general population in some respects, as they often share many of the same social, biological, and environmental exposures as the survivors. However, using siblings as the reference group should yield conservative estimates of the absolute excess risk. Furthermore, the shared uncontrolled confounders may also be considered a strength.”

2. The first half of the discussion, as currently written, reads a bit like a restatement of the results. Rather than merely repeatedly stating that this study is the first of its kind, it could be helpful for the authors to more clearly describe how this study's findings fit within existing theoretical frameworks to understand childhood cancer survivors' long-term lifestyle-related risks.

Response: With this study describing the associations between both individual and combined lifestyle factors and 16 outcomes, we have made an effort to summarize these results and put them into context, both in relation to previous studies in childhood cancer survivors and studies on benefits of healthy lifestyle in the general population. Furthermore, we appreciate the suggestion of describing how the study's findings fit with existing theoretical frameworks. We consider the results to fit very well both with the ongoing frameworks of both precision health and premature aging in childhood cancer survivors. Therefore, we have added the following sentences to the discussion:

“Furthermore, with chronic health conditions considered a phenotype of accelerated aging in childhood cancer survivors, it is possible that these result signal potential reversal of this accelerated aging by healthy lifestyle, as has been suggested in cross-sectional studies. This potential for mitigation of treatment-related accelerated aging, combined with the higher AERs in survivors with unhealthy lifestyles, show potential for lifestyle interventions as part of precision health initiatives in childhood cancer survivors.”

3. On the top of page 8, the authors state, ...may be more cost-effective in childhood survivors than in the general population if we assume a similar effectiveness for behavior change...” However, it is likely not appropriate to assume that healthy lifestyle interventions designed for healthy populations would have the same effectiveness for pediatric cancer survivors. I encourage the authors to consider rewording.

Response: Thank you for pointing this out. We have now rephrased the sentence to the following:

“Our results suggest that the same behavior change will reduce the risk for diabetes, heart failure, valvular heart disease, arrhythmia, anxiety, depression, and QoL impairments more in childhood cancer survivors than in the general population.”

4. As written, the discussion seems to imply that researchers have yet to develop healthy lifestyle interventions that specifically target adolescents/young adults or young adults who are pediatric cancer survivors. However, several such interventions exist and continue to be tested in current literature. It could be helpful for the authors to specifically (albeit briefly) either provide some examples of such interventions and/or describe what the limitations of such interventions are (and how the current study’s findings suggest the existing interventions should be strengthened). For example, although some recent healthy lifestyle interventions for AYA/YA pediatric cancer survivors have begun focusing on limiting attrition and improving participant engagement, many previous interventions demonstrated relatively high attrition.

Response: We did not intend to imply that no interventions have been developed. With the broad scope of our study, reporting observational associations between lifestyle and a variety of chronic health conditions, we feel a review of ongoing and recently performed intervention trials would be beyond the scope of our manuscript. We hope that our results can support decisions to fund more trials aimed at showing results on clinically relevant outcomes from lifestyle interventions. To address the point of the reviewer, we have clarified our sentence by adding “further”:

“Hence, effective lifestyle interventions aimed at long-term change need to be further developed and tested in childhood cancer survivors.”

5. On page 8, the authors suggest that many previous healthy lifestyle interventions have not achieved “fully healthy lifestyles.” What does this phrase mean, operationally?

Response: We appreciate pointing the need for clarification. We have now revised the sentence:

“As lifestyle interventions may only result in a partial shift towards healthier behaviors, it is promising that the AERs were larger in survivors than sibling controls even for moderately unhealthy lifestyle.”

6. On page 7, lines 175-176, it is unclear what the authors mean when they state, “Our results now substantiate previous reports that have suggested associations between a healthy lifestyle and reduced risk for late mortality.” Wouldn’t late mortality (as opposed to death at a young age) be a good thing? Hasn’t most extant literature shown that healthy lifestyles do, in fact, tend to delay mortality into later adulthood?

Response: In childhood cancer research, “late mortality” refers to mortality occurring after the initial cancer-related mortality, typically ≥ 5 years. See e.g. the article by Dixon et al. in Lancet “Specific causes of excess late mortality and association with modifiable risk factors among survivors of childhood cancer: a report from the Childhood Cancer Survivor Study cohort”. However, with the confusion this caused, we changed the wording from “late mortality” to “excess mortality.”

7. In the limitations section, the authors rightfully report that BMI is not an appropriate proxy for healthy eating behaviors because BMI is impacted by physical activity, biological diathesis, and eating behaviors. However, BMI could also be impacted by cancer treatment-related late-effects, like metabolic changes. Because of the nature of the study sample, I encourage the authors to reflect the potential impact of cancer treatment (or even tumor location in the case of some brain tumors) on BMI.

Response: We agree that unfavorable BMI may be caused by childhood cancer treatment. We added the following sentence to the limitations section:

“While unfavorable BMI may also be related to childhood cancer treatment exposures, our analyses were adjusted for treatment exposures so the observed BMI association should not be confounded by treatment exposure.”

8. On page 8, I recommend breaking the text into multiple, shorter paragraphs to reduce cognitive load for the reader. For example, the authors might consider beginning new paragraphs at “However, BMI reflects both diet...” and at “Both the exposures and outcomes in our study were self-reported...”

Response: We appreciate this suggestion and it is also in line with the editorial comments. We have now revised the paragraphs accordingly.

9. The authors state that causality can be implied by their study’s longitudinal design; however, because causality can only be inferred from experimental design, it is overstated to say that even a large-sample, longitudinal study can imply causality.

Response: We agree that experimental designs provide the strongest basis for causal inference. However, following the framework proposed by Bradford Hill, causality should not be viewed as a dichotomous property that is either present or absent, but rather as something that can be inferred to varying degrees based on multiple lines of evidence. Poorly performed experimental studies can provide even less proof of causality than a very well-performed observational study. Longitudinal observational studies with appropriate temporal ordering, large sample sizes, and careful control of confounding can, therefore, contribute meaningfully to causal inference, particularly when supported

by biological plausibility, consistency with prior evidence from other populations, and dose-response patterns. To clarify this, we have revised the sentence to the following:

“While causality cannot be claimed in an observational study, it should be noted that longitudinal observational studies such as ours with appropriate temporal ordering, large sample sizes, and careful control of confounding can contribute meaningfully to causal inference, particularly when supported by biological plausibility, consistency with prior evidence from other populations, and dose-response patterns.”